# Structurally distinct telomere-binding proteins in *Ustilago maydis* execute non-overlapping functions in telomere replication, recombination, and protection

Eun Young Yu[1], Syed S. Zahid[1], Swapna Ganduri[1], Jeanette H. Sutherland[1], Min Hsu[1], William K. Holloman[1] & Neal F. Lue [1,2 ✉]

Duplex telomere binding proteins exhibit considerable structural and functional diversity in fungi. Herein we interrogate the activities and functions of two Myb-containing, duplex telomere repeat-binding factors in *Ustilago maydis*, a basidiomycete that is evolutionarily distant from the standard fungi. These two telomere-binding proteins, *Um*Tay1 and *Um*Trf2, despite having distinct domain structures, exhibit comparable affinities and sequence specificity for the canonical telomere repeats. *Um*Tay1 specializes in promoting telomere replication and an ALT-like pathway, most likely by modulating the helicase activity of Blm. *Um*Trf2, in contrast, is critical for telomere protection; transcriptional repression of *Umtrf2* leads to severe growth defects and profound telomere aberrations. Comparative analysis of *Um*Tay1 homologs in different phyla reveals broad functional diversity for this protein family and provides a case study for how DNA-binding proteins can acquire and lose functions at various chromosomal locations. Our findings also point to stimulatory effect of telomere protein on ALT in *Ustilago maydis* that may be conserved in other systems.

[1] Department of Microbiology & Immunology, W. R. Hearst Microbiology Research Center, Weill Cornell Medicine, 1300 York Avenue, New York, NY 10065, USA. [2] Sandra and Edward Meyer Cancer Center, Weill Cornell Medicine, 1300 York Avenue, New York, NY 10065, USA. ✉email: nflue@med.cornell.edu

Eukaryotic chromosome ends, or telomeres, harbor repetitive DNA sequences that mediate the assembly of a special, dynamic nucleoprotein structure that is crucial for genome stability. This dynamic structure stabilizes chromosome ends by suppressing aberrant repair events at the termini[1,2] and by promoting the retention and replenishment of telomere DNA through successive rounds of replication. Special mechanisms are required to retain and replenish telomere DNA owing to (i) the propensity of telomere DNA to form barriers that block replication forks[3]; and (ii) the "end-replication" problem that prevents the complete synthesis of lagging strand duplex[4]. Both the protective and DNA maintenance functions of telomeres are mediated by specific proteins within the dynamic assembly, often through direct physical interactions between telomere proteins and functionally relevant downstream targets.

Within the dynamic telomere nucleoprotein assembly, duplex telomere repeat-binding factors are known to execute crucial functions including telomere length regulation, telomere protection and telomere replication. Studies in diverse organisms have revealed considerable structural and functional variabilities in these duplex-binding factors. In mammals, the two major duplex telomere-binding proteins, Telomere Repeat-Binding Factor 1 and 2 (TRF1 and TRF2), are each part of the shelterin complex that collectively protects chromosome ends and suppresses the DNA damage response. While TRF1 and TRF2 are structurally similar in having an N-terminal TRF homology dimerization domain (TRFH) and a C-terminal Myb domain, each protein serves distinct functions in telomere regulation. TRF2, in particular, plays a key role in telomere protection by inhibiting telomere fusions and the DNA damage response[2]. TRF1, on the other hand, promotes efficient telomere replication by recruiting BLM helicase to unwind replication barriers[5]. (TRF2 also physically associates with BLM, but the functional significance of this is unclear[6].) In the fission yeast *Schizosaccharomyces pombe*, a single TRF1/2-like protein named Taz1 alone performs multiple functions in telomere protection and replication, although it is unclear how the replication function is executed[7,8]. Interestingly, *S. pombe* contains two other Myb-containing proteins capable of binding telomere repeat-like sequences. One of these, named Tbf1, resembles mammalian TRF1/2 according to sequence alignment in the TRFH and Myb domains (in fact, more so than does Taz1), but exhibits no compelling telomere functions[9]. The other protein, named *Sp*Teb1, bears two consecutive Myb motifs away from the C-terminus and evidently functions primarily as a transcription factor[10]. Interestingly, the *Sp*Teb1 homolog in the budding yeast *Yarrowia lipolytica* (named Tay1, a well-studied member of this protein family) is reported to be a key telomere-binding protein and plays a critical role in telomere maintenance and protection[11]. While the evolutionary basis for the structural and functional divergence of duplex telomere-binding proteins is not fully understood, a likely driver is the irregularity of the fission and budding yeast telomere repeats (e.g., TTAC$_{0-1}$A$_{0-1}$C$_{0-1}$G$_{1-9}$ for *S. pombe* and TG$_{1-3}$ for *Saccharomyces cerevisiae*) and its deviation from the canonical GGGTTA repeat found in mammals and all major eukaryotic branches. (For a more detailed discussion of telomere sequence divergence in budding and fission yeasts and the corresponding changes in duplex telomere-binding proteins, see Sepsiova et al. and Steinberg-Neifach and Lue[12,13].)

A potentially informative model with which to address mechanistic and evolutionary issues related to duplex telomere-binding proteins is the basidiomycete *Ustilago maydis*. This model was originally developed by Robin Holliday to investigate recombinational repair[14]. We were initially attracted to this model—in comparison to other fungal models—because (1) *Ustilago maydis* carries a telomere repeat unit that is identical to the mammalian repeat[15]; (2) the recombination and repair

machinery in this organism bears greater resemblances to those in mammals[16]; and (3) repair proteins in this organism (e.g., Rad51, Blm, and Brh2) promote telomere replication and telomere recombination just like their mammalian ortholog counterparts[17–19]. Accordingly, we have taken advantage of *U. maydis* to examine the mechanisms and regulation of repair proteins at telomeres[17–20]. Interestingly, with respect to duplex telomere-binding proteins, our initial database query (as well as an earlier bioinformatics screen) revealed in the *U. maydis* genome just one potential duplex telomere-binding protein that is structurally similar to *Sp*Teb1 and *Yl*Tay1, and that we and others had named *Um*Trf1[12,20,21]. We showed in a published report that this protein has high affinity and sequence specificity for the cognate telomere repeats, and that it interacts directly with Blm helicase, just like mammalian TRF1/2[20]. To underscore that this protein is structurally distinct from mammalian TRFs, we will hereafter refer to it as *Um*Tay1. While our initial findings suggest that *Um*Tay1 could be the main duplex telomere repeat factor in *U. maydis*, subsequent analysis of *tay1Δ* mutants revealed little telomere deprotection phenotypes (see below), suggesting that the true telomere capping protein remains to be identified. Indeed, Tomaska and colleagues have noted the existence of a TRF/TBF1-like gene in *U. maydis* with unknown function and potential TRFH and Myb domains[12], which we will hereafter designate as *Um*Trf2 (Uniprot ID: A0A0D1E1Z3). Thus, *U. maydis* apparently harbors two structurally distinct, Myb-containing proteins that may mediate distinct or overlapping functions in telomere protection and maintenance. Because Basidiomycetes occupy a more basal position in fungal phylogeny than the ascomycete budding and fission yeasts, studies of duplex telomere-binding proteins in this lineage may offer insights on the origin of these protein families in fungi and the division of labor between them vis-à-vis other organisms.

Herein we present our structural and functional analysis of *Um*Tay1 and *Um*Trf2. We showed that both proteins, despite their unusual structural features (see Results), recognize the canonical telomere repeat GGGTTA/TAACCC with high affinity and sequence specificity. Trf2 but not Tay1 is critical for telomere protection. Trf2 is evidently essential, and transcriptional repression of this gene results in profound telomere aberrations, accumulation of ssDNA, and elevated levels of extrachromosomal telomere repeats (ECTR). Tay1, on the other hand, contributes to telomere replication and positively stimulates an ALT-like pathway of telomere recombination, most likely by interacting with Blm and controlling its helicase activity. This functional division of labor is unlike that observed in metazoans or in budding/fission yeasts. Cross comparison of our results with those from other fungal and non-fungal systems highlights the potential of structurally distinct telomere proteins to execute similar functions in different lineages. Based on the accumulated data, we propose that the emergence of the Tay1 family of telomere proteins in fungi may have provided added flexibility to the fungal telomere system to adapt to drastic changes in telomere sequences in budding and fission yeasts.

## Results

**Tay1 binds GGGTTA/TAACCC repeats through tandem Myb motifs.** As noted before, *Um*Tay1 is similar to *Sp*Teb1 and *Yl*Tay1 in harboring two consecutive Myb motifs that manifest substantial similarities to the Myb motif of TRF proteins (Supplementary Fig. 1a). However, unlike the other homologs, *Um*Tay1 possesses a much longer C-terminal extension, rendering the protein twice as large as *Sp*Teb1 and *Yl*Tay1. To confirm the DNA-binding activity of the Myb domain and investigate the potential role of the C-terminus, we purified an N-terminal 274

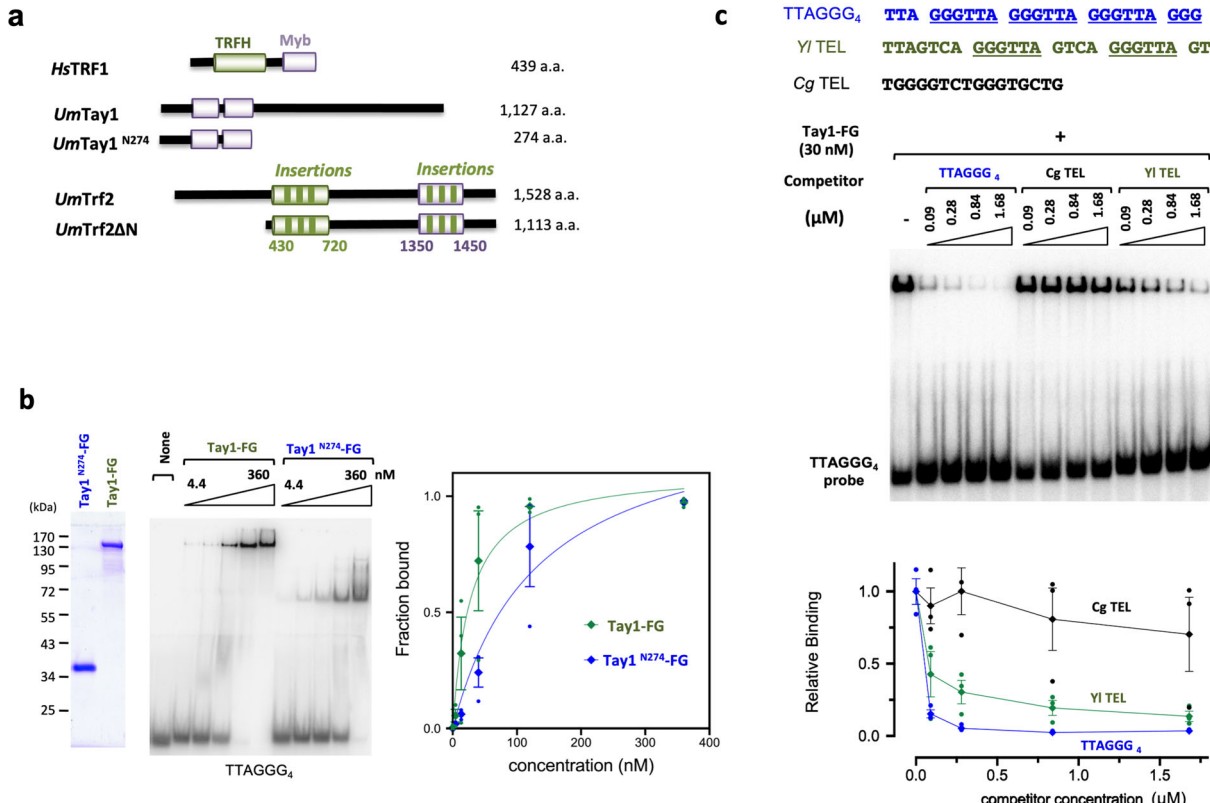

**Fig. 1 The DNA-binding activity of *Ustilago maydis* Tay1. a** Schematic illustrations of the domain structures of the *U. maydis* Tay1 and Trf2 proteins analyzed in this study, as well as that of human TRF1, a prototypical member of the TRF/TBF1 protein family. The TRFH and Myb domains are shown in green and purple, respectively. All the *U. maydis* recombinant proteins examined in this work also contain a C-terminal FLAG tag (as indicated by the designations Tay1-FG and Tay1$^{N274}$-FG), which is not illustrated. **b** EMSA analysis of the DNA-binding activities of Tay1-FG and Tay1$^{N274}$-FG (TayF1 and Tay1$^{N274}$ carrying a C-terminal FLAG tag). The assays were performed using a series of progressive, 3-fold increases in protein concentrations ranging from 4.4 to 360 nM. A representative set of assays is shown on the left and the quantitation (average ± SEM, n = 3 independent experiments) shown on the right. Previous investigations of *Sp*Teb1 revealed $K_d$s that range from 20 nM to >1 μM, possibly owing to the different DNA substrates used in these studies[12,50]. Our estimated $K_d$s for Tay1 and Tay1$^{N274}$ (27 and 130 nM, respectively) are broadly in line with these earlier analyses. **c** The DNA-binding specificity of Tay1-FG was examined using three different double-stranded competitor oligonucleotides. The sequences of the competitors are shown on top with the canonical GGGTTA repeat unit underlined—only the G-strand sequences are displayed. *Cg*TEL and *Yl*TEL are based on the telomere repeat of *Candida glabrata* and *Yarrowia lipolytica*, respectively. A representative series of assays is shown and the quantified results (average ± SEM, n = 3 independent experiments) plotted on the bottom.

amino acid (a.a.) fragment (spanning the duplicated Myb motifs and named Tay1$^{N274}$) and compared its DNA-binding properties to those of the full length protein (Fig. 1a). In a titration analysis, Tay1 exhibited ~4-fold higher affinity binding to telomere DNA than Tay1$^{N274}$; the estimated $K_d$s of these proteins for the TTAGGG$_4$ probe are 27 and 130 nM, respectively (Fig. 1b). We also assessed the sequence-specificity of Tay1 using competitor oligonucleotides harboring different telomere sequences. The *Yl*TEL oligo (derived from the *Y. lipolytica* telomere sequence), in which the canonical 6-bp repeat unit (GGGTTA) is 4-nt apart, was slightly less active than the cognate 6-bp repeat (TTAGGG$_4$) (Fig. 1c). In contrast, the *Cg*TEL oligo (derived from the *Candida glabrata* telomere sequence), another G-rich repeat that lacks the GGGTTA sequence, was unable to compete. This binding preference is similar to that reported for *Y. lipolytica* Tay1, specifically with regard to the strong preference for DNA with canonical repeats[11,12], and indicates that *Um*Tay1 has the requisite DNA-binding properties to act specifically at *U. maydis* telomeres.

**Trf2 is also a high affinity telomere repeat-binding factor.** In addition to *Um*Tay1, Sepsiova et al. noted the potential existence of a TRF/TBF1-like protein (Uniprot ID: A0A0D1E1Z3), which, being 1528 a.a. long, is considerably larger than a typical TRF/

TBF1 ortholog such as human TRF1 (Fig. 1a). We named this gene *Umtrf2*, and performed additional bioinformatic analysis by identifying and characterizing closely related homologs in the Pucciniomycotina and Ustilaginomycotina subphyla (Supplementary Fig. 1b, c). Like *Um*Trf2, these fungal proteins are ~1.5 to 3 times the sizes of mammalian TRFs. Nevertheless, multiple sequence alignment of these proteins against metazoan TRF orthologs revealed the existence of both a central TRFH domain and a C-terminal Myb domain, precisely as predicted for a true TRF ortholog (Supplementary Fig. 1b, c). Notably, the fungal proteins carry multiple insertions in both their TRFH and Myb domains (vis-à-vis the metazoan proteins), thus accounting in part for their larger sizes. Notwithstanding these insertions, structural considerations suggest that the fungal protein domains may retain the function of TRFH and Myb in dimerization and DNA-binding. For example, even though the Myb motif of *Um*Trf2 harbors three short insertions (Fig. 1a and Supplementary Fig. 1c), these insertions map to regions of the motif that are distal to the DNA-binding surface as judged by sequence alignment and structure of the human TRF1–DNA complex (Fig. 2a). Accordingly, they are not predicted to disrupt *Um*Trf2-DNA binding. Notably, whereas insertion 1 is quite variable in length (2–15 a.a.), insertions 2 and 3 are relatively invariant in length

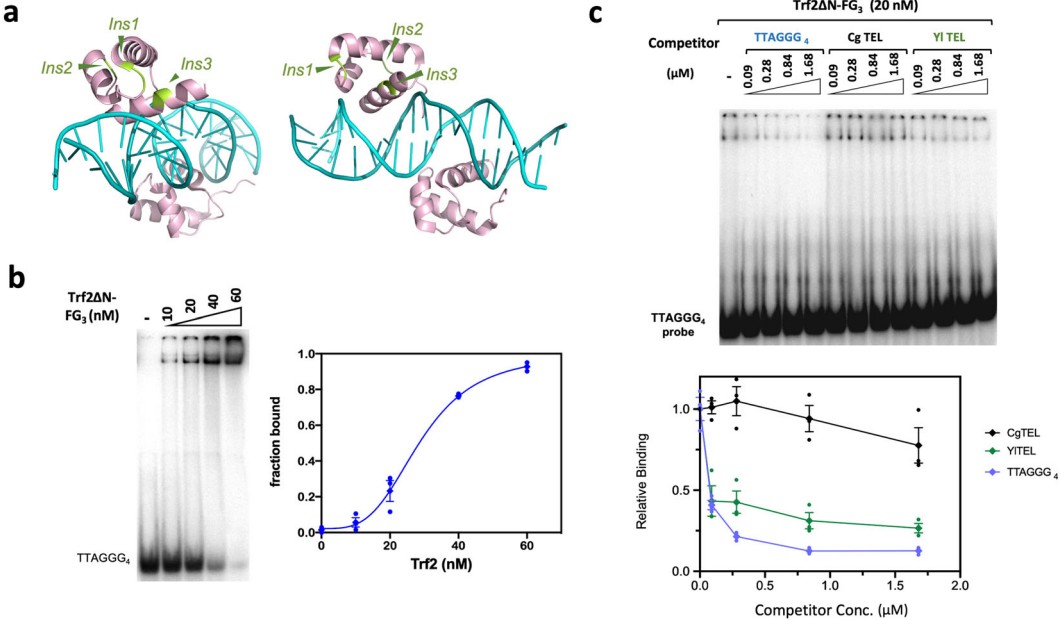

**Fig. 2 The DNA-binding activity of *Ustilago maydis* Trf2. a** The structure of human TRF1–DNA complex is used to illustrate the locations of three insertions within the Myb domain of *U. maydis* Trf2. The positions where the insertions occur are shown in light green. Note that these are all positioned away from the DNA–protein interface. **b** EMSA analysis was performed using varying concentrations of Trf2ΔN-FG$_3$ and double-stranded telomere oligonucleotides. A representative series of assays is shown and the quantified results (average ± SEM, *n* = 3 independent experiments) plotted on the bottom. **c** The DNA-binding specificity of Trf2 was examined using the same competitor oligonucleotides as in Fig. 1c. A representative series of assays is shown and the calculated results (average ± SEM, *n* = 3 independent experiments) plotted on the bottom.

and contain well-conserved residues that may serve important functions other than DNA-binding.

To test the DNA-binding activity of *Um*Trf2 directly, we attempted to purify the full length protein but were unsuccessful. Because the long N-terminal extension of this protein is probably unstructured[22] and poorly conserved even among fungi, we tested and succeeded in purifying a truncated derivative that is missing the first 415 amino acids (Trf2ΔN-FG$_3$). Purified Trf2ΔN-FG$_3$ binds with high affinity to double-stranded TTAGGG$_4$ ($K_d$ of ~20 nM) but not the G-strand alone (Fig. 2b and Supplementary Fig. 2a). We then purified and analyzed the putative TRFH region and Myb domain of Trf2 separately (amino acid 416-1349 and 1350-1528, respectively), and found that as predicted, only the Myb domain exhibited low affinity binding to the telomeric probe (Supplementary Fig. 2b).

Further analysis indicates that the two *U. maydis* telomere-binding proteins manifest quite similar DNA-binding properties. For example, oligo competition experiments indicate that just like Tay1, Trf2 strongly prefers DNAs that contain the canonical 6-bp repeat (Fig. 2c). In addition, the two proteins have nearly indistinguishable minimal DNA target sizes of ~2 to 3 repeats (Supplementary Fig. 3a, b). The TTAGGG$_{2.5}$ probe, which harbors 2 copies of the GGGTTA repeats (the target of a single Myb motif according to structural analysis[23]), was bound with lower affinity by both Tay1 and Trf2 in comparison to TTAGGG$_{3.5}$ and TTAGGG$_4$, which have 3 copies of GGGTTA. On the other hand, TTAGGG$_{2.0}$, with just 1 copy of GGGTTA, did not support any detectable binding (Supplementary Fig. 3). Thus, despite the structural differences between Tay1 and Trf2, the two *U. maydis* proteins evidently recognize telomere repeat DNA using very similar mechanisms. This notion is also consistent with conservation of putative DNA-binding residues between *Um*Tay1 Myb1, UmTay1 Myb2, *Um*Trf2 Myb, and metazoan TRF Myb domains (Supplementary Fig. 1a, c).

**Tay1 is required for telomere replication but not protection**. To investigate the telomeric functions of Tay1 in vivo, we analyzed the phenotypes of multiple, independently derived *tay1Δ* clones. These clones were found to exhibit similar growth rates as the parental strain (FB1), and manifest no evidence of senescence following prolonged passages, suggesting that Tay1 is not essential for either telomere protection or telomerase function. However, telomere restriction fragment (TRF) Southern analysis revealed substantially shortened but stably maintained telomeres in the *tay1Δ* mutants (Fig. 3a). The shortening was evident at streak 3 (corresponding to ~100 generation after the construction of the mutants), but did not worsen at later time points. Indeed, both wild type and *tay1Δ* telomeres grew slightly during continuous sub-culturing (Fig. 3a), a phenomenon that has been noted earlier[18]. The telomere phenotypes of *tay1Δ* clones are reminiscent of DNA repair mutants and point to telomere replication as the main function of this protein[18,20]. Previous characterization of the telomere replication mutants (e.g., *blmΔ* and *rad51Δ*) revealed preferential loss of long telomeres[24]. This feature is also true of *tay1Δ* telomeres: in STELA analysis of UT4-bearing telomeres, the longest telomere tracts found in *tay1Δ* were ~300 bp shorter than those in the wild type strain, whereas the shortest tracts were similar in size (Fig. 3b). (UT4 is a previously characterized subtelomeric element in *U. maydis*[15,24].) We also considered the alternative possibility that Tay1 may promote telomerase function rather than telomere replication. To test this, we engineered conditional mutants of *trt1* and *tay1* by placing these two genes under the control of the arabinose-dependent promoter *crg1*, such that their expression is permissive in YPA and repressed in YPD. After 4 streaks (~100 generations) in YPD, the *trt1^crg1 tay1^crg1* double mutant manifested greater telomere loss than either *trt1^crg1* or *tay1^crg1*, indicating that the two genes do not act in the same pathway (Supplementary Fig. 4).

In contrast to the telomere length defects observed in Southern and STELA analysis, in-gel hybridization assays did not uncover

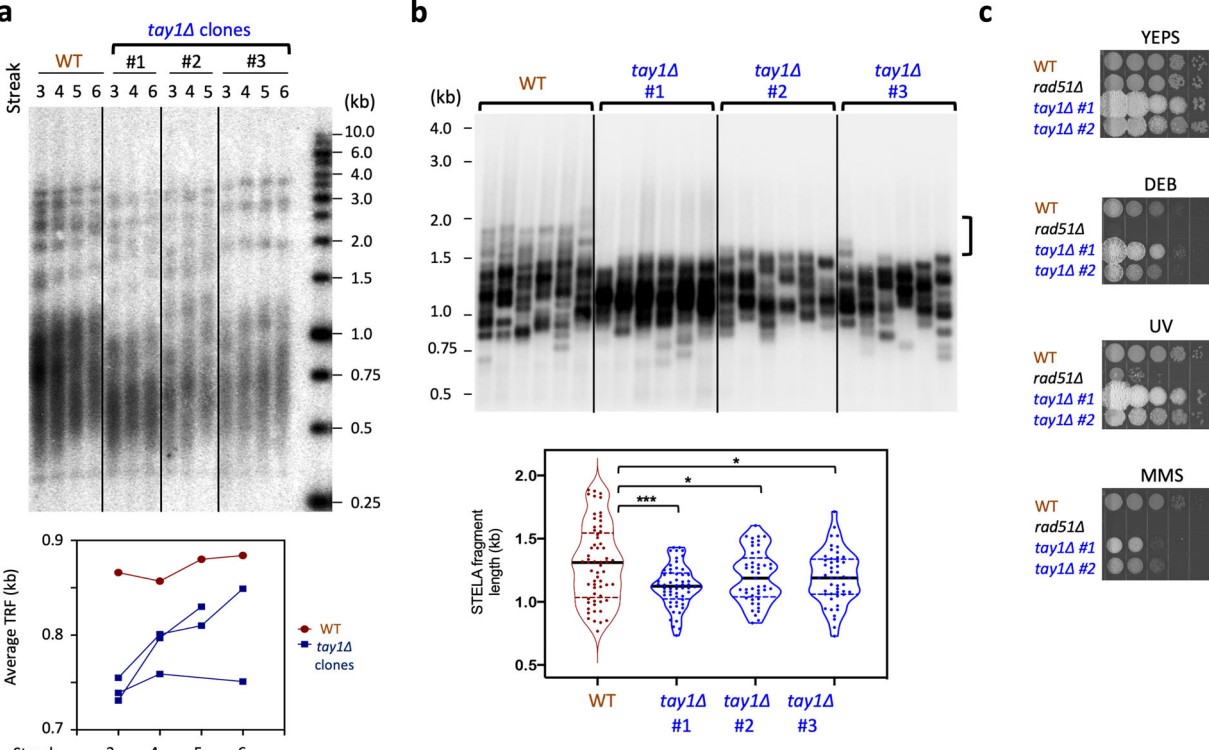

**Fig. 3 *Ustilago maydis* Tay1 is required for optimal telomere replication, and plays a minor role in DNA damage response. a** Chromosomal DNAs were prepared from wild type and three independent *tay1Δ* transformants following the indicated number of streaks. Following *Pst*I cleavage, the DNAs were subjected to TRF Southern analysis. The phosphor scan of the blot is shown along with the plot of average telomere length for each sample. **b** Chromosomal DNAs from wild type and *tay1Δ* strains were subjected to STELA. Following ligation to the Teltail oligonucleotide, UT4/5-containing telomeres were amplified using a UT4/5-specific primer, and detected using a UT4/5 probe. Representative assays are shown on the top, with the vertical bracket to the right highlighting the long telomere fragments missing from the *tay1Δ* samples. The composite STELA profiles (collected from $n = 6$ independent STELA PCR reactions) are displayed as violin plots (with means and quartiles) on the bottom. **c** Ten-fold serial dilutions of the indicated strains were spotted on YPD containing the indicated clastogens (0.02% MMS Methyl methanesulfonate; 0.01% DEB, 1,2,3,4-diepoxybutane) or subjected to UV irradiation (120 J/m²), and their growth assessed after 2–3 days.

obvious changes in ssDNA at *tay1Δ* telomeres. In addition, we did not detect any C-circles, which are extrachromosomal circular telomere species that often accumulate when de-protected telomeres become recombinogenic[19,25]. Thus, the predominant telomere function of *Um*Tay1 is evidently limited to telomere replication. We also assessed the function of *tay1* in DNA repair, and found that *tay1Δ* did not manifest greater susceptibility to various DNA-damaging agents than the parental strain, except for a moderate sensitivity to MMS (Fig. 3c, bottom panel).

**Tay1 binds to and controls the activity of Blm helicase.** We have previously shown that just like mammalian TRF1 and TRF2, *Um*Tay1 physically interacts with the conspecific Blm helicase[20]. Given this interaction and the similarity between the telomere phenotypes of *tay1Δ* and *blmΔ*, we hypothesize that Tay1 promotes telomere replication by controlling Blm activity. To test this possibility biochemically, we analyzed the effects of Tay1 on Blm helicase activity in vitro (Fig. 4a). Two double-stranded DNA substrates with 3′ overhangs, one carrying a telomeric sequence and the other a non-telomeric sequence, were used to assess the helicase activity of Blm, a 3′ to 5′ helicase[26]. Notably, the untreated telomeric substrate migrated as two distinct species in native gels, most likely owing to G-rich overhang-mediated dimer formation, especially given the propensity of G-rich sequences to form higher order structures such as G-quadruplexes (Fig. 4b, leftmost lane; Supplementary Fig. 5, 6th lane). As expected, the addition of Blm resulted in the unwinding of duplex DNA into

ssDNA in an ATP-dependent manner (Supplementary Fig. 5). Interestingly, the effects of Tay1 on unwinding were substrate sequence-dependent: on telomeric DNA, Tay1 inhibited DNA unwinding by Blm by up to 70%, whereas on non-telomeric DNA, the effect was stimulatory by up to 2.5 fold (Fig. 4a, b; Supplementary Fig. 6a). The inhibitory effect of full length Tay1 can be recapitulated by the N-terminal Myb domain (Tay1 [N274]), suggesting that high affinity DNA-binding by Tay1 may be sufficient to block the unwinding activity of Blm (Supplementary Fig 6b). In contrast, Tay1 [N274] is unable to stimulate unwinding on the non-telomeric substrate, implying that the C-terminal regions of Tay1 contribute to this activity, possibly through protein–protein interactions (Supplementary Fig. 6a). It should be noted that Blm was included at 12.5 and 25 μM for the non-telomeric and telomeric assays, respectively, in order to accentuate the stimulatory and inhibitory effects of Tay1.

To examine functional interactions between *Um*Tay1 and Blm in vivo, we generated *blmΔ* and *tay1Δ blmΔ* mutants in the same strain background as *tay1Δ* and compared their telomere phenotypes. All three mutants manifested stably shortened telomeres, consistent with a telomere replication defect (Fig. 4c). Notably, as judged by the loss of long telomeres in both telomere Southern and STELA analysis, the defect is more severe in *tay1Δ blmΔ* than in either single mutant (Fig. 4c, Supplementary Fig. 7). The exacerbated phenotype of *tay1Δ blmΔ* vis-à-vis *tay1Δ* and *blmΔ* is most notable in the upper range of STELA size distribution; the longest fragments in *tay1Δ* and *blmΔ* (~1.3 to 1.5 kb) are mostly absent in *tay1Δ blm1Δ* (Supplementary Fig. 7). These results suggest that the two genes do

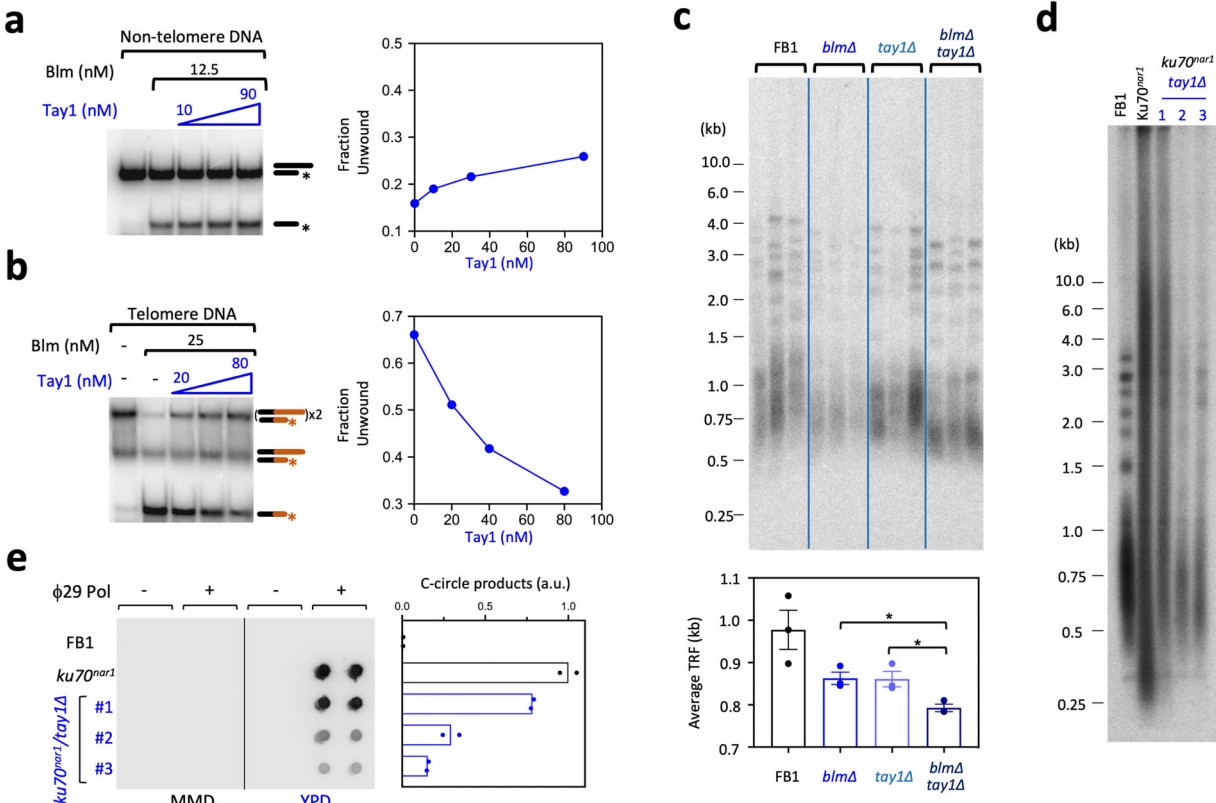

**Fig. 4 Tay1 modulates Blm helicase activity and functionally resembles Blm in regulating telomere replication and recombination. a** The helicase activity of Blm (12.5 nM) on a non-telomeric substrate was examined in the presence of increasing Tay1 concentrations (0, 10, 30, and 90 nM). The quantified results are shown on the right. **b** The helicase activity of Blm (25 nM) on a telomeric DNA-containing substrate was examined in the presence of increasing Tay1 concentrations (0, 20, 40, and 80 nM). The quantified results are shown on the right. A higher concentration of Blm was chosen for these assays than those in (**a**) in order to accentuate the inhibitory effect of Tay1. **c** Chromosomal DNAs were isolated from three independent cultures of wild type or mutant strains and subjected to telomere restriction fragment analysis. The phosphor scan of the blot is shown along with the plot of telomere lengths (average ± SEM, $n = 3$ biologically independent samples) for each strain. The mutant strains were all passaged for ~75 generations before the analysis. **d** Chromosomal DNAs were isolated from the indicated strains (grown in YPD), and subjected to telomere restriction fragment Southern analysis. The *ku70^{nar1} tay1Δ* samples 1–3 are from three independently constructed strains. **e** Chromosomal DNAs were isolated from the indicated strains with the given growth condition, and subjected to duplicate C-circle assays. The scan of the blot is shown on the left and the quantification (means; $n = 2$ experimental replicates; normalized against the average value of *ku70^{nar1}* samples) plotted on the right.

not act exclusively in a single pathway in promoting telomere replication. Indeed, we had previously detected an interaction between *U. maydis* Blm and Pot1, raising the possibility that Blm may be regulated by multiple telomere proteins[20]. Likewise, it is possible that Tay1 may interact with another factor(s) that contributes to telomere replication. However, with respect to potential interacting partners, we note that Tay1 does not appear to bind Rad51/Brh2, another DNA repair complex implicated in telomere replication[18] (Supplementary Fig. 8).

Another previously established function of Blm at telomeres is to trigger an ALT-like pathway when telomeres are de-protected. More specifically, when the *ku70/80* complex is transcriptionally repressed in *U. maydis* (e.g., in the *ku70^{nar1}* mutant), telomeres become highly aberrant and manifest a collection of abnormalities that resemble those in ALT cancer cells, including telomere length heterogeneity, ssDNA on the C-strand, and high levels of C-circles[17,19]. All these phenotypes are completely suppressed in *ku70^{nar1} blmΔ*, implying an essential role for *blm* in triggering the ALT pathway[19]. Given the ability of Tay1 to regulate Blm activity, we proceeded to generate *ku70^{nar1} tay1Δ* mutants and analyzed their telomeres. Notably, all three independently generated *ku70^{nar1} tay1Δ* exhibited milder telomere length abnormalities than *ku70^{nar1}* when *ku70* expression was repressed. In particular, telomere length distributions became less heterogeneous, and the levels of C-circles were substantially reduced

in *ku70^{nar1} tay1Δ* relative to that in *ku70^{nar1}* (Fig. 4d, e). The levels of C-strand ssDNA in the double mutants were also lower than that in the *ku70^{nar1}* single mutant (Supplementary Fig. 9a). Consistent with the mild telomere aberrations, the *ku70^{nar1} tay1Δ* double mutants exhibited very little growth defects and did not manifest the enlarged and elongated morphology characteristic of G2/M arrest (Supplementary Fig. 9b, c). It is worth noting that one of the *ku70^{nar1} tay1Δ* mutants (#1) consistently manifested milder suppression of the *ku70^{nar1}* defects; another mutation might have been introduced during the construction of this strain. Collectively, our results indicate that *tay1Δ* partially phenocopies *blmΔ* in the context of Ku70 depletion, and support the notion that Tay1 positively regulates Blm in activating the ALT pathway. The residual telomere aberrations in the *ku70^{nar1} tay1Δ* mutants suggest, however, that Tay1 is not absolutely essential for Blm function in this pathway.

**Trf2 is crucial for telomere protection**. The lack of an obvious telomere de-protection phenotype in *tay1Δ* mutants suggests that *Um*Trf2 may be the key telomere protection factor. To test this idea, we attempted to generate null mutants by gene replacement. Multiple attempts were unsuccessful, suggesting that the gene may be essential. We then constructed transcriptionally conditional mutants of *trf2* (named *trf2^{crg1}*) in which the endogenous *trf2* promoter is replaced by the *crg1* promoter[27]. As noted before,

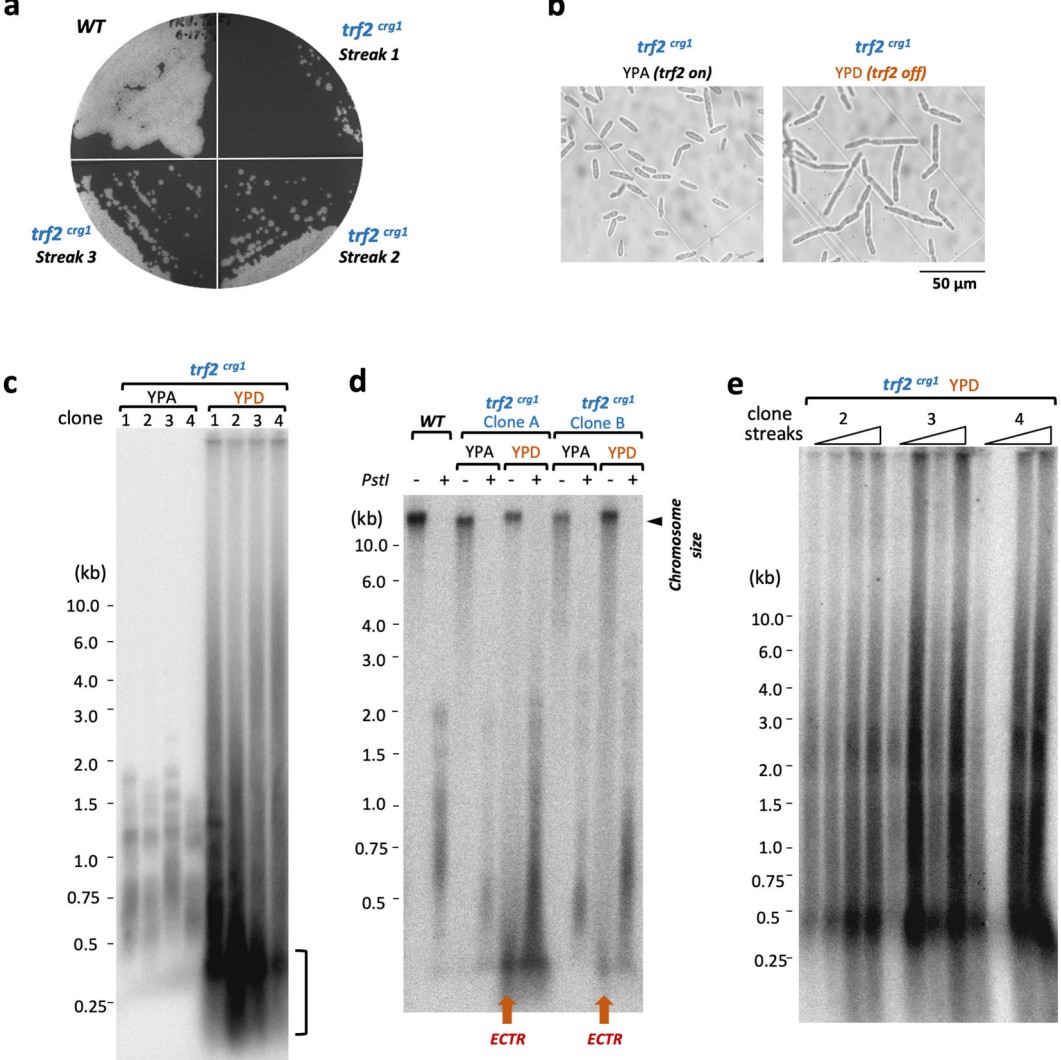

**Fig. 5 *Ustilago maydis* Trf2 is critical for cell proliferation and telomere protection. a** A *trf2crg1* clone was passaged on YPD (a medium that represses *trf2* transcription), and the growth of the clone at successive streaks assessed along with the parental strain. The plates were incubated at 30 °C for 3 days. **b** A *trf2crg1* clone was grown in liquid YPA or YPD, and examined under the microscope. **c** DNAs from multiple, independently constructed *trf2crg1* mutants that had been grown in YPA or YPD were subjected to TRF Southern analysis. **d** Untreated or *Pst*I-digested chromosomal DNAs were fractionated on agarose gel, transferred to nylon membrane, and then subjected to Southern hybridization using a TTAGGG$_{82}$ probe. The short telomere fragments that can be detected in the absence of *Pst*I digestion are highlighted as extra-chromosomal telomere repeats (ECTR). **e** Multiple independent *trf2crg1* clones were passaged on YPD plates by repeated re-streaking. DNAs from liquid culture out-growth of the resulting colonies were subjected to telomere restriction fragment Southern analysis.

this promoter permits *trf2* expression in YPA but not in YPD. Indeed, multiple, independently derived *trf2crg1* mutants grew poorly in YPD (albeit with some differences in growth rate), indicating that this protein is required for optimal cell proliferation (Fig. 5a). The mutants also manifest elongated morphology that is consistent with cell cycle defects (Fig. 5b). TRF Southern analysis revealed, in the *trf2*-repressed clones, grossly abnormal telomeres characterized by (1) high overall telomere repeat content, (2) telomere size heterogeneity, and (3) high levels of extremely short telomere fragments (Fig. 5c, marked by a vertical bracket). Notably, a large fraction of telomere DNA in the mutant probably exists as ECTR, given that short telomere fragments were detected even in the absence of restriction enzyme digestion (Fig. 5d). All three aberrant telomere features noted above were evident in multiple, independently generated *trf2crg1* clones, and during serial passages in YPD, implying that Trf2 is critical for telomere protection (Fig. 5e).

The role of Trf2 in telomere protection is also supported by analysis of single-stranded telomere DNA through in-gel hybridization. In particular, low levels of G-strand ssDNA (single-strand DNA) were detected in *trf2*-expressing clones, consistent with normal telomeres harboring G-strand, 3′-overhangs (Fig. 6a). In contrast, in *trf2*-repressed clones, G-strand ssDNA was reduced whereas C-strand ssDNA became greatly elevated (Fig. 6a). Moreover, the amounts of C-circles, another marker of abnormal telomere recombination, were also increased in these clones (Fig. 6b). Notably, these ssDNA and C-circle aberrations are also found in *ku70*- or *ku80*-repressed mutants, suggesting that the abnormal telomere processing reactions in the *trf2* and *ku*-deficient mutants may share some mechanistic resemblance. However, there may be differences as well, given that the levels of C-circles in the *ku70nar1* are more than 10-fold higher than those in *trf2crg1* (cf. Figs. 4e, 6b, note that the two sets of assays were performed

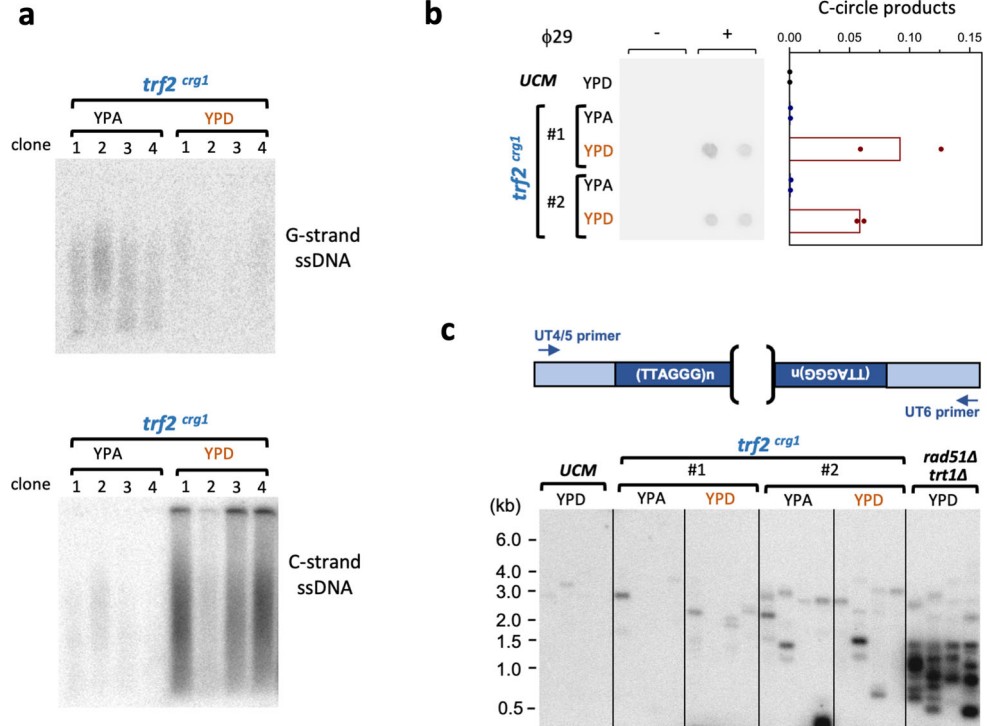

**Fig. 6 Ustilago maydis Trf2 suppresses multiple telomere aberrations. a** DNAs from multiple, independently constructed *trf2crg1* strains grown in YPA or YPD were assayed for the levels of unpaired G-strand and C-strand telomere DNA by in-gel hybridization. **b** The indicated chromosomal DNA preparations were subjected to duplicate C-circle assays. The scan of the blot is shown on the left and the quantification (means; *n* = 2 experimental replicates; normalized against the average value of *ku70nar1* samples in Fig. 4e) plotted on the right. Note that these assays were performed concurrently with those in Fig. 4e and quantified from the same blot. **c** The DNAs from the indicated strains grown in the given media were subjected to telomere fusion analysis.

concurrently, analyzed on the same blot, and quantified using the same scale).

To further examine the protective function of *trf2*, we developed a PCR-based assay to assess telomere–telomere (T–T) fusions between UT4/5 and UT6-bearing telomeres (Supplementary Fig. 10a). Notably, the vast majority of PCR products generated in these assays hybridized to both a UT4/5 probe and a UT6 probe, as predicted for products derived from authentic fusions (Supplementary Fig. 10a). Cloning and sequencing of a 1.1 kb product further confirmed the presence of UT4/5 and UT6 sequences on the two ends of this fragment (Supplementary Fig. 10b). Applying the fusion assay to the *U. maydis trt1Δ rad51Δ* double mutant revealed high levels of T–T fusions, which are consistent with previous studies of similar mutants in other organisms[28,29] (Supplementary Fig. 10a). In contrast, the levels of fusions in the *trf2*-repressed clones are only slightly elevated, suggesting that the defective telomeres in these cells are not as prone to non-homologous end joining (Fig. 6c).

## Discussion

Fungi have been a rich source of discovery for telomere biology. Nevertheless, except for a few standard and atypical organisms in the ascomycete phylum, the telomere machinery in different fungal lineages remains poorly understood. We have sought to characterize telomere regulation in *Ustilago maydis* as a starting basis for understanding telomeres in Basidiomycetes. Because of the resemblance of the repair machinery in *U. maydis* to that in metazoans, this fungus could also serve as a useful model for the interplay between the telomere and repair machinery. In this study, we focused on the structural, functional, and mechanistic features of two double-strand telomere-binding proteins in *U. maydis*, named *Um*Tay1 and *Um*Trf2. Notably, we showed that

both proteins, despite their unusual and quite different structural features, recognize the canonical telomere repeat GGGTTA/ TAACCC with very similar affinity and sequence specificity. Furthermore, the two proteins mediate quite distinct telomere regulatory functions, with Tay1 contributing primarily to telomere replication and telomere recombination, and Trf2 being the chief telomere protection factor. These observations have interesting mechanistic and evolutionary ramifications, as discussed below.

The DNA-binding mechanisms of TRF/TBF1 protein family have been studied extensively. High resolution structures of the Myb domain–DNA complexes revealed crucial contacts between the two Myb protomers (head-to-tail arrangement) and consecutive GGG/CCC triplets in the telomere DNA targets[23]. In addition, single particle EM analysis of full length TRF1 suggests that the TRFH dimerization domain helps to position the Myb domains in a relative orientation that is conducive to DNA binding[30]. In contrast, small angle x-ray scattering (SAXS) envelope of TRF2 dimer revealed an extended conformation that leaves the two Myb protomers far apart[31]. It seems likely that, given its similar structural organization and similar DNA-binding specificity, *Um*Trf2 would recognize telomere repeats in the same fashion as mammalian TRF1 and TRF2, i.e., through specific interactions between Myb protomers and two neighboring GGG/ CCC triplets. One notable feature of *Um*Trf2 is the large linker between its TRFH and Myb domains (~600 a.a. compared to ~100 a.a. in human TRF1 and ~200 a.a. in human TRF2), a feature that could provide additional flexibility to *Um*Trf2 in positioning its two Myb protomers relative to the DNA target. This idea is consistent with the ability of *Um*Trf2 to bind *U. maydis* and *Y. lipolytica* telomere sequences with similar affinities, despite the differences in gaps between the consecutive GGG/ CCC triplets (3 bp for the *U. maydis* repeat and 7 bp for the *Y. lipolytica* repeat).

Less is known about the DNA-binding mechanism of the Tay1 protein family, which are characterized by two tandem Myb motifs separated by ~20 a.a. (or much longer in some homologs such as those from *Coccidioides posadasii* or *Uncinocarpus reesei*)[11,20]. While no high resolution structural information is available for this protein family, the similarities in the DNA target size and sequence preference exhibited by *Um*Tay1 and *Um*Trf2 suggest that Tay1-like proteins may position their two Myb motifs on telomere DNA in the same way as TRF/TBF1 proteins. Notably, structural considerations suggest that the linker between the two Myb motifs in Tay1 (minimally 20 a.a.) is adequate to span the distance between the C-terminus of one Myb protomer and the N-terminus of the other protomer as judged by the crystal structure of the TRF1–DNA complex (30–40 Å depending on the combination). The long linker between the Myb motifs may also account for the ability of *Um*Tay1 to bind *Y. lipolytica* telomeres, in which the GGGTTA core repeat is separated from each other by four nucleotides. Even though the two Myb motifs are expected to be further apart along the DNA axis, they would lie on the same side of the DNA double helix, thus reducing the spatial separation that needs to be spanned by the protein linker. Therefore, we propose that a Tay1 monomer might be sufficient for high affinity and sequence-specific binding to telomere repeats. Indeed, our preliminary co-expression/pull down experiments did not reveal detectable interactions between differently tagged *Um*Tay1 (Supplementary Fig. 11), suggesting that the protein does not form a stable dimer. However, we cannot exclude the possibility of DNA-induced oligomerization, which was suggested by EM analysis of *Y. lipolytica* Tay1–DNA complex[11].

The Tay1 protein family, which encompasses *Um*Tay1, provides an interesting case study in the plasticity of the telomere machinery (Fig. 7). Tay1 homologs appear to be confined to fungi, and database analysis suggests that they emerged prior to the divergence of the Dikarya clade, since family members can be identified in both Basidiomycete and Ascomycetes subphyla (e.g., homologs in *Aspergillus* (Pezizomycotina), *Schizosaccharomyces* (Taphrinomycotina), *Puccinia* (Pucciniomycotina), and *Yarrowia* (Saccharomycotina)), but not in more basal branches[11,12]. Interestingly, this gene was evidently lost in late branching lineages of Saccharomycotina, given the absence of any discernible homologs in, e.g., *Saccharomyces* and *Candida* spp. Indeed, not only is the existence of Tay1 homologs in Basidiomycetes and Ascomycetes variable, but also their participation in telomere regulation. Our results indicate that at least in *U. maydis*

(and possibly in other Basidiomycetes), this protein promotes telomere replication and regulates telomere recombination but is not essential for telomere protection or maintenance. In contrast, *S. pombe* Tay1 functions primarily as a transcription factor[10], and *Y. lipolytica* Tay1 seems to be essential for telomere maintenance or protection; *Y. lipolytica tay1Δ* haploid strains could not be generated, and a *Tay1/tay1Δ* diploid strain exhibited dramatic telomere shortening in comparison to the wild type diploid strain[11].

The remarkable plasticity of Tay1 function may be rationalized by considering (1) the relatively recent emergence of the Tay1 protein family in fungi and (2) the drastic alterations in telomere repeat sequence in budding and fission yeasts. One can surmise that prior to the emergence of Tay1 homologs, the functions of duplex telomere-binding protein would be exclusively executed by a TRF/TBF1-like protein in the common ancestor of Basidiomycetes and Ascomycetes. Once Tay1 evolved to bind telomeres, it could have been enlisted to take over an existing function from the ancestral TRF/TBF1, or it might evolve a new function in telomere regulation. The role of *Um*Tay1 in telomere replication is consistent with either scenario. As a non-essential player at telomeres, the ancient Tay1 would have been relatively unconstrained in adopting other functions, as in the case of SpTay1, which regulates transcription rather than telomeres. On the other hand, the different fate of Tay1 in Saccharomycotina is more easily rationalized with respect to the telomere sequence divergence. In *Y. lipolytica*, the retention of a telomere repeat that contains the canonical GGGTTA sequence enables Tay1 to be telomere-bound and to function in telomere maintenance/protection—though it is possible that another TRF/TBF1-like protein named *Yl*Tbf1 may also act at telomeres[11] (Fig. 7). In the later branching lineages of Saccharomycotina yeasts (e.g., *Saccharomyces* and *Candida*), the complete absence of the GGGTTA sequence renders telomeres refractory to binding by either Tay1 or TRF/TBF1, thus necessitating the utilization of another Myb-containing protein (Rap1) with a more relaxed sequence specificity to bind and protect telomeres[13]. While TBF1 in these organisms might have been retained because of its roles in transcription and subtelomere regulation[32–34], Tay1 was probably lost due to the lack of any critical function. Altogether, the emergence of Tay1 in fungi and its subsequent loss in specific lineages provide interesting illustrations of functional acquisition and loss in evolution by a new protein family. The ability of this

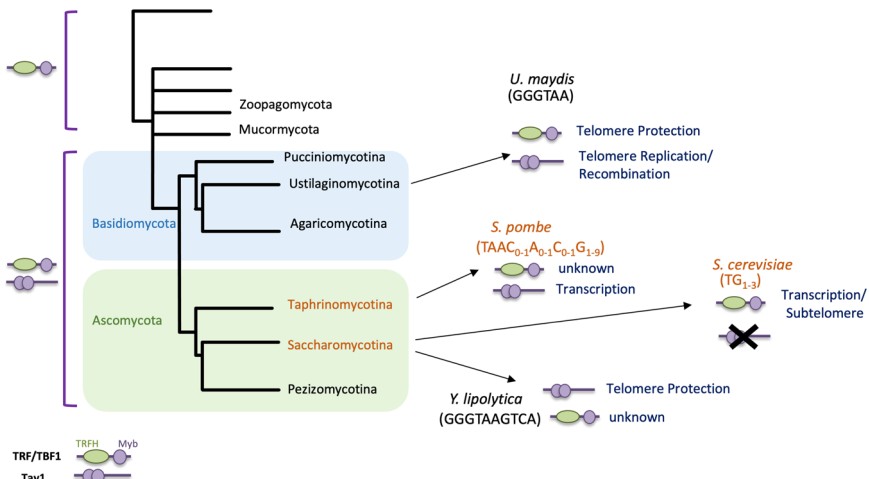

**Fig. 7 Functional plasticity of the Tay1 protein family.** A simplified fungal phylogeny is shown along with the distribution of the TRF/TBF1 and Tay1 protein family in the phylogeny. The tree is adapted from a previous study[51]. The phyla in which the telomere repeats deviate from the canonical GGGTTA sequence are indicated in red. The existence and functions of TRF/TBF1 and Tay1 homologs in selected organisms are also illustrated.

protein family members to evolve different telomeric and non-telomeric functions may have provided added flexibility to fungi to meet the challenges posed by telomere sequence alterations. Indeed, beyond TRF/TBF1 and Tay1, the plasticity of duplex telomere-binding proteins is further illustrated by Rap1 and Taz1, two other Myb/Sant-containing proteins that emerged later to serve as the major TBPs in budding and fission yeasts, respectively[12,13].

Another key finding in this study is the demonstration of a stimulatory function for UmTay1 in the ALT-like pathway in U. maydis. This is to our knowledge the first example of a telomere protein exerting a positive regulatory function in ALT-related pathways and could have relevance for similar pathways in mammalian systems. Both mammalian TRF1 and TRF2 have been shown to bind and possibly regulate the cognate BLM, and BLM is strongly implicated in promoting ALT and telomere BIR (break-induced replication)[5,6,35,36]. While these observations hint at a positive regulatory role for mammalian TRFs in ALT, no direct evidence has been reported. Experimental interrogation of this idea is likely to be challenging: To wit, given the roles of TRF1 and TRF2 in telomere integrity and telomere protection, studies that address the interplay between TRF1/TRF2 and BLM in ALT will require appropriate separation-of-function mutants. In this regard, the rather limited telomere function of tay1 in U. maydis allowed us to address the issue more easily through the use of tay1Δ null mutations.

The current work also points to potential complications in studies that investigated ALT mechanisms through the use of TRF1-FokI, a fusion protein that induces telomere-specific breaks[37,38]. The DNA synthesis/repair triggered by the induction of TRF1-FokI has been proposed to mimic the natural ALT pathway. However, if TRF1-FokI retains the ability to recruit or activate BLM, the breaks generated by this fusion protein might be preferentially processed by BLM, and preferentially channeled toward specific repair pathways rather than those utilized by endogenous ALT substrates. Given that there are multiple ALT pathways with distinct factor requirements[39–41], the roles of telomere proteins in regulating these pathways and in modulating the outcomes of BIR-like DNA synthesis at telomeres would appear to warrant further investigation.

## Methods

**Ustilago maydis strains and growth conditions**. Standard protocols were employed for the genetic manipulation of U. maydis[42–44]. All U. maydis strains used in this study were haploid and were derived from the UCM350 or FB1 background[43,45] and are listed in Supplementary Table S1.

The ku70nar1 mutant harbors a conditional ku70 under the control of the nitrate-dependent promoter from nar1[17]. Accordingly, the ku70 gene in this strain is expressed in the MMD medium and repressed in YPD. The tay1Δ mutants were generated by replacing the entire tay1 open reading frame in FB1 with a cassette expressing resistance to hygromycin (Hyg^R) through homologous recombination[18,43,46]. Briefly, the plasmid harboring the tay1 disruption cassette with the HYG^R marker (named pUTay1-Hyg) was constructed through Gibson Assembly (NEB) of four overlapping PCR fragments, which were generated using appropriate primers (Supplementary Table S2) and either genomic DNA or plasmid pUMar261 as templates. The four PCR fragments linked by Gibson assembly include: (i) 5′ UTR of tay1 connected to the 5′ end of the hygromycin ORF, (ii) the complete hygromycin ORF flanked by short segments from the 5′ and 3′UTR of tay1, (iii) 3′ end of the hygromycin ORF linked to 3′ UTR of tay1, and (iv) the vector sequence from pUMar261 flanked by short segments from the 5′ and 3′UTR of tay1. The disruption cassette was released from the vector by cleaving pUTay1-Hyg with EcoRI and BamHI, and used for transformation. The ku70nar1 tay1Δ mutants were constructed from the tay1Δ mutant using the pUKU70nar1 plasmid[17].

The trf2crg1 strain was constructed by integrating a linearized pRU12-trf2 plasmid containing fragments that match (1) 700 bp of the trf2 promoter and (2) the first 700 bp of the trf2 ORF into the trf2 genomic locus in UCM350. Briefly, the two trf2 fragments were generated by PCR using appropriate oligos with flanking restriction enzyme sites (Supplementary Table S2), and then cloned together in between the NdeI and EcoRI sites of pRU12[27]. Correct recombination of XbaI treated pRU12-trf2 into the trf2 locus results in the insertion of the CBX^R marker

and the crg1 promoter precisely adjacent to and upstream of the trf2 ATG initiation codon. Following transformation, the correct mutant strains were selected on YPA supplemented with carboxin (4 μg/ml) and confirmed by PCR. The strains were grown in YPA (1% yeast extract, 2% peptone, 1.5% arabinose) or YPD (1% yeast extract, 2% peptone, 2% dextrose) to induce or repress trf2 expression, respectively. The tay1crg1 strain was made using the same approach, i.e., by constructing the equivalent pRU12-tay1 plasmid and integrating XbaI-linearized pRU12-Tay1 into the tay1 genomic locus. The trt1crg1 strain was similarly engineered except that the trt1 fragments (corresponding to an upstream region and a 5′ portion of ORF) were cloned in between the NdeI and BamHI site of pRU11-hyg, and the transformants were selected on YPA supplemented with hygromycin (150 μg/ml).

The genotype of each mutant was confirmed by PCR reactions that verified both the disruption of the endogenous locus and the integration of the cassette into the correct chromosomal locus. Notably, despite some phenotypic variations among the tay1Δ, trf2crg1, and ku70nar1 tay1Δ mutants, they all display the expected PCR products from the disrupted loci, and are devoid of the wild type genes (Supplementary Fig. 12).

**Telomere length and structural analyses**. Southern analysis of telomere restriction fragments (TRF) was performed using DNA treated with PstI or AluI plus RsaI[18,20]. The blots were hybridized to a cloned restriction fragment containing 82 copies of the TTAGGG repeats. The in-gel hybridization assays for detecting ssDNA on the G- and C-strand were performed using a published protocol[17]. After the gels were dried without denaturation, the ss telomere DNAs were detected using labeled TTAGGG$_4$ and CCCTAA$_4$ oligonucleotides (Supplementary Table 2) as the probes, and hybridization was performed in the Church Mix at 45 °C. STELA for telomeres bearing UT4 subtelomeric elements was performed essentially as reported before[24]. After scanning of the blots, the length distributions of STELA fragments were analyzed using TeSLA software[47]. To detect telomere–telomere fusions, chromosomal DNA was subjected to PCR using two subtelomeric primers (UT4-2116-F and UT6-2210-F) that are extended by DNA polymerase towards the chromosome ends (Fig. 6c). The PCR reactions (20 μl) contained 1x Failsafe PreMix H, 0.1 μM each of UT4-2116-F and UT6-2210-F, and 2 U Failsafe™ polymerase, and the thermocycling consisted of 36 cycles of 94 °C for 30 s, 63 °C for 30 s, and 72 °C for 2 min. The fusion PCR products were detected by Southern analysis using an UT6 subtelomeric probe[24]. C-circle assays were carried out by the rolling circle amplification method[20,25]. Briefly, genomic DNAs (10 ng) were added to 20 μl reaction containing 0.2 mg/ml BSA, 0.1% Tween, 1 mM each of dATP, dGTP, and dTTP, 1× phi29 Buffer and 5 U NxGen® phi29 DNA polymerase (Lucigen). The reactions were performed at 30 °C for 16 h, and then the DNA polymerase inactivated at 65 °C for 20 min. The products were dot-blotted onto a 2× SSC-soaked nylon membrane (Hybond-N, GE Healthcare, Inc.). Following UV-cross-linking, the G-strand products of C-circle assays were detected through hybridized at ~55 °C to $^{32}$P-labeled CCCTAA$_8$ probe.

**Expression and purification of Tay1, Trf2, and Blm**. To express Tay1, Tay1$^{N274}$, Trf2-TRFH(416-1349), Trf2-Myb(1350-1528), and Blm, we generated open reading frames with C-terminal FLAG tags from genomic DNA by PCR amplification using suitable primer pairs (Supplementary Table 2). The PCR fragments were cloned into the pSMT3 vector to enable the expression of fusion proteins. For the N-terminally truncated Trf2 (amino acid 416 to 1528), the PCR fragment was inserted into a modified pSMT3 vector carrying a FG$_3$ tag downstream of the cloning site. BL21 codon plus strains bearing the expression plasmids were grown in LB supplemented with kanamycin and induced by IPTG[48]. Each fusion protein was purified by (i) Ni-NTA chromatography, (ii) ULP1 cleavage and, and (iii) anti-FLAG affinity chromatography[49].

**EMSA assays**. For DNA-binding assays, purified proteins were incubated with 1–10 nM P$^{32}$-labeled probe and 0.5 μg/μl BSA in binding buffer (25 mM HEPES-KOH, pH 7.5, 5% glycerol, 3 mM MgCl$_2$, 0.1 mM EDTA, 1 mM DTT) at 22 °C for 15 min, and then subjected to electrophoresis at 0.8 V/cm through a 6% non-denaturing polyacrylamide gel in 0.2x TBE, followed by PhosphorImager analysis. To estimate K$_d$s, binding reactions were performed in triplicates, and the resulting data fitted to the one site binding (hyperbola) model in GraphPad Prism (GraphPad Software Inc.).

**Helicase assays**. The assays (modified from[6]) were performed in 20 μl of helicase buffer (20 mM Tris.HCl, pH 7.5, 2 mM MgCl$_2$, 0.1 mg/ml BSA, 1 mM DTT, 2 mM ATP) containing 0.4 nM labeled dsDNA and indicated amounts of Blm-FG, Tay1-FG, and Tay1$^{N274}$-FG. After 20 min of incubation at 35 °C, the reactions were terminated by the addition of 4 μl stop solution (30% glycerol, 50 mM EDTA, 0.9% SDS, 0.1% BPB and 0.1% xylene cyanol) and 20 ng of unlabeled bottom strand oligo (as trap to prevent reannealing of unwound ssDNA). The products were subjected to electrophoresis at 1 V/cm through 10% polyacrylamide gels (acrylamide : bis-acrylamide :: 29 : 1) containing 1x TBE, and then quantified by PhosphorImager analysis. The ratio of labeled DNA in single-stranded form (bottom species in the gel) to total DNA (all the labeled species) at the end of the reaction is taken as the fraction unwound by Blm helicase.

**Statistics and reproducibility**. Student's t tests were performed using GraphPad Prism 8, and the p-values were calculated and indicated as follows: *$p < 0.05$; **$p < 0.01$; ***$p < 0.001$. Each experiment was repeated two or three times to ensure reproducibility. All comparisons between different mutants were conducted using two or three independently generated mutants of the same genotypes.

## Data availability

The data generated and/or analyzed during the current study are included within the paper and Supplementary Information or available from the corresponding author upon reasonable request. The original, uncropped gel and blot images for Figs. 1–6 are included in the Supplementary Information file as Supplementary Fig. 13. The source data for all the plots in the paper are included in the Supplementary Data 1 file.

## Code availability

The source code for TeSLA can be downloaded at the following link: https://www.nature.com/articles/s41467-017-01291-z#Sec25[47].

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

## Acknowledgements

We thank José Pérez-Martin (CSIC, Spain) for plasmids, Qingwen Zhou for purified Rad51 and Brh2/Dss1, Lubomir Tomaska for pointing out the existence of *Um*Tbf1/Trf2, and members of our laboratories for discussion. This work was supported by a pilot project grant from the Sandra and Edward Meyer Cancer Center (N.F.L. and W.K.H.), NSF MCB-1817331, and NIH GM 107287 (N.F.L.).

## Author contributions

E.Y.Y., W.K.H., and N.F.L. conceived the study. E.Y.Y., S.Z., and N.F.L. designed and executed the experiments with contributions from S.G., J.S., and M.H.; E.Y.Y. and N.F.L. interpreted the data and wrote the manuscript with advice from S.Z. and W.K.H.

## Competing interests

The authors declare no competing interests.
