## [Transparent Peer Review File · Communications Biology]

Reviewers' Comments:

Reviewer #1:

Remarks to the Author:

The paper by Yu et al. provides evidence that *Ustilago maydis* employs two distinct telomere-binding proteins (TBPs) that play complementary functions in telomere maintenance in this organism. While UmTrf1 seems to play a primary role in telomere replication, UmTrf2 is involved in telomere protection.

The results of the study are very interesting in terms of both (1) mechanism of telomere maintenance in *U. maydis*; and (2) evolution of telomeres. The experiments are very well designed and executed and represent a complex mix of *in vitro* and *in vivo* assays resulting in a nice, highly entertaining story.

I believe that the paper is an important addition to the field and I have only a few comments:

1. p.3: the sentence „TRF1, on the other hand, ...“ indicates that only TRF1 recruits BLM helicase to telomeres. As shown by Lillard-Wetherell et al. (Hum. Mol. Genet., 2004) and as mentioned by the authors in the other parts of the manuscript, both TRF1 and TRF2 associate with Blm.
2. p.3: first mention of *Schizosaccharomyces pombe* should be in unabbreviated form
3. p.3: sentence „The other protein...“ the reference to Fig. 1 does not seem appropriate.
4. p.4: SpTay1 is not an official name of the protein, I suggest to call it SpTeb1 as indicated in PomBase database: <https://www.pombase.org/gene/SPAC13G7.10>
5. The terminology of the various types of TBPs is somewhat confusing. UmTrf1 (and SpTeb1) are Tay1-like (this is clear), UmTrf2 is described as TRF and Tbf1-like. Is it really true that yeast Tbf1 proteins (such as ScTbf1, CaTbf1 and SpTbf1) are homologues of human TRF1/TRF2, or do they represent another group of TBPs? Do these proteins contain TRFH domain homologous to mammalian TRFs? Even Taz1 of *S. pombe*, originally thought to be a homologue of TRF1/TRF2 was shown to be structurally unrelated protein (<https://www.ncbi.nlm.nih.gov/pmc/articles/PMC4493283/>) so calling UmTrf2 as TRF1/TRF2-like is a little far-fetched.
6. The discussion implies that Tbf1 protein was lost in *Yarrowia lipolytica*. However, this is not the case. As originally described by Kramara et al. (2010), there is a gene YALI0B18018 encoding a homologue of ScTbf1, whose function at telomeres of *Y. lipolytica* was so far not carefully assessed. Thus the text as well as a scheme presented on Figure 7 should be corrected.
7. p.5 and Figure 1 legend: CgTEL oligo: it should be indicated that it is derived from the telomeric repeat of *Candida glabrata*; YITEL --- from *Yarrowia lipolytica*
8. p.5: The sentence: „In addition...“: a reference to Fig. 1A would be more appropriate.
9. TRF analyses presented on Figures 3-5: In some cases, the differences in lengths of TRFs do not seem as obvious as claimed by the authors. Therefore, the lengths of TRFs should be subjected to quantitative analysis. In addition, the intensity of the signals varies between the lanes. As no loading control(s) are presented, it is not clear, if this variation is due to different amounts of total DNA or different relative amount of telomeric sequences.
10. Figure 3C: the mutant *trf1Δ* seems to exhibit a different growth on the three upper plates. Is it reproducible?
11. Figure 4: Is there a reason why the authors used different concentrations of both Blm and Trf1 in experiments presented as Fig. 4A and 4B respectively. In legend to Figure 4D it should be indicated, what represent the numbers 1-3 above the lanes (independent clones? Different passages?)
12. Figure 5B should contain a scale bar.
13. Although it was described by de Sena-Tomas et al. (2015) the authors should briefly (e.g. in Methods section) explain what is the basis regulation of *ku70^{nar1}* allele.
14. The authors claim that division of labor between UmTrf1 and UmTrf2 are unprecedented, but

mammalian TRF1 and TRF2 (even though they, in contrast to UmTrf1 and UmTrf2 are homologues) also exhibit division of labor at telomeres.

15. In the discussion, the affinities of UmTrf1 and UmTrf2 to TTAGGG and TTAGGG-like repeats should be compared with K_D obtained for other Tay1-like and Tbf1-like proteins, respectively, in previously published studies.

16. p. 10: the sentence „The Tay1 protein family...“: there should be a reference to Figure 7 (not Figure 6).

17. The discussion on evolution of yeast telomere-binding proteins (visualized on Figure 7) could be extended also to evolution of Rap1 and Taz1 as a major TBPs in some phylogenetic branches and compare their current views with the scenarios previously proposed by the authors' and other laboratories.

Reviewer #2:

Remarks to the Author:

The experiments reported in this manuscript describe telomere maintenance functions of two fungal Myb-containing proteins that have been found based on sequence analogy with human TRF1 and TRF2 Myb domains. The experimentation is performed mainly in vitro with purified bacteria-expressed variants of proteins: UmTrf1 was expressed in full length but UmTrf2 was expressed without N-terminal 415 AA residues.

This appears to be the second report that examines how fungal proteins containing Myb domain interact with telomeric DNA. As far as I know, the first study describing telomeric proteins in fungi has been published by the Tomaska laboratory (Kramara J, et al. Tay1 protein, a novel telomere binding factor from *Yarrowia lipolytica*. J Biol Chem, 2010).

The key findings of recent study are that UmTrf1 is required for telomere maintenance, UmTrf1 modulates Blm helicase and UmTrf2 is essential for cell survival and sustainable telomere protection. I am afraid that the conclusions drawn are insufficiently supported by the data. The measured data and conclusions could have been more robust if additional data points and replicates are measured and presented (see below).

Given the level of detail provided, methodology information and validity of statistical analyses are insufficient, especially in regards to the ability of a researcher to reproduce the work.

Taken together, the major conceptual drawback for me is the fact that the authors performed most of the experiments with truncated variant of UmTrf2 lacking N-terminal part. In case of human TRF2, N-terminal part is responsible for additional DNA binding modes. Thus, if only the truncated variant is investigated, we can miss important functional contributions of N-terminal part of UmTrf2.

The manuscript contains good initial data, but in present form, it needs more carefully conducted experiments with full-length proteins to the extent that significantly diminishes my enthusiasm for the paper.

Therefore, I suggest rejecting the manuscript in present form.

The authors could consider kindly suggestions that might improve their experimental concept and manuscript.

1. Full-length UmTrf2 should be expressed using a yeast expression system.
2. EMSA assays should be carried out with at least nine data points and in three replicates to obtain reliable and robust conclusions.
3. Statistical analyses and fitting functions should be described clearly. It is not obvious how the authors determined K_D values.
4. Methods should be provided with all details to the level that enables a researcher to reproduce the

work.

For example:

- a. "C-circle assays were carried out by the rolling circle amplification method (22), with modifications that were also previously described (19)." should be described precisely.
- b. Electrophoresis assays should be provided with electric field E values in V/cm.

Reviewer #3:

Remarks to the Author:

In this manuscript, Yu et al., characterize two distinct duplex telomere repeat binding proteins in the fungus *Ustilago maydis*, which shares the mammalian telomeric repeat, in contrast to other studied fungi in the ascomycetes phyla. They present findings related to the ability of these proteins, *UmTrf1* and *UmTrf2*, which are referred to hereafter as simply Trf1 and Trf2, to bind various telomeric repeat containing substrates despite the presence of multiple short insertions within the TRFH- and myb-like domains. They additionally present data suggesting Trf1 is required for maintenance of long telomere tracts and stimulates ALT-like telomere recombination, activities that may be achieved through modulating the activity of Blm. They further demonstrate that, in contrast to Trf1, Trf2 is essential and is involved in telomere end protection (inhibition of ECTR formation and C-strand SS DNA). The finding that Trf1, like Blm, suppresses the ALT-like telomere phenotype in *ku70^{nar1}* cells is quite striking, as is the impact of conditional suppression of Trf2 on telomere length and integrity. Lastly, the development of an assay to detect telomere-telomere fusions in *U. maydis* is a potential technical strength (see below). The authors use these findings to ponder in the discussion the plasticity of the Tay1 protein family, to which Trf1 belongs. While the findings and discussion are interesting, there are several experimental details that need to be addressed:

Major issues

1. There appears to be some variability of the impact of *trf1Δ* on telomere length. In Fig. 3A, the bulk TRF lengths in clone #3 appear similar to WT. The lengths in clone 1 seem uniformly decreased rather than exhibiting just shortening of the longer telomeres. Moreover, the STELA analysis is unclear. Typically, multiple independent reactions are prepared from DNA isolated from a single culture. I assume this is the case here as well. Thus, while the STELA analysis of the single WT and *trf1Δ* clones looks convincing for a possible selective effect of loss of Trf1 on long telomeres, an analysis of multiple clones is needed.
2. Data demonstrating telomerase-dependent or -independent shortening or defects in replication of duplex telomeric repeats is needed to conclude that Trf1 contributes to telomere replication.
3. The description of the telomeric probes in Supp. Fig 2 might be incorrect. For example, the sequence of TTAGGG₄ is different than in Fig. 1C and has a repeat of GGGTAA. The sequences of the other probes are also in question.

4. The interpretation of the Supp. Fig. 2 needs reconsideration. First, what is the basis for the templates designating TTAGGG₂ and TTAGGG₄ having two and four repeats, respectively, and that there is any functional relevance to a half repeat or a repeat split across the ends of a probe? An alternative interpretation of the data in Supp. Fig. 2 is that Trf1-TG has equivalent binding to the TTAGGG₄ and TTAGGG_{3.5} probes, which each has 3 complete repeats; binds less to the TTAGGG_{2.5} probe which has just 2 complete repeats; and does not bind the TTAGGG₂ probe which has just a single repeat.
5. The growth properties of the *trf1*Δ clones in Fig. 3C appears strikingly different than the growth of the WT and *rad51*Δ. This is particularly true on YEPS and DEB and after UV treatment. Moreover, MMS has a rather dramatic effect on this property, suggesting the strains have more than a slight sensitivity.

6. Fig. 4B and Supp. Fig.4B How are the fractions unwound calculated given the two distinct bands in the absence of protein added? How is it that the telomeric DNA substrates form a stable dimeric structure if the overhangs are not complementary? Why would the “dimeric” molecule be preferentially unwound by BLM (compare e.g., top and middle bands in 4B lane 1 with comparable bands in lane 2; same type of analysis could be done for Supp. Fig. 4)? Lastly, how reproducible are the effects observed in these helicase activity assays (quantification appears to be done on single experiments).
7. What is the most rapidly migrating band in the telomeric DNA lanes in Suppl. Fig 3? The putative single stranded labeled molecule is designated to the right. Does that make sense?
8. The median and 3rd interquartile values for *blmΔ* and *trf1Δ blm1Δ* do not appear different in Suppl. Fig. 5B, so these data do not support the conclusion of a more severe effect in the double mutant. Additionally, there appears to be a low of the shorter fragments in the STELA experiment but not in the TRF analysis. Could it be that the shorter fragments detected by Southern are not amplified in the STELA?
9. The TRF, c-circle, and C strand ssDNA analyses comparing the *ku70^{nar1}* and *trf1Δ ku70^{nar1}* strains shows clear differences, however, the suppression of the telomere length heterogeneity and c-circles is quite variable (Figs. 4D, 4E, Suppl. Fig 7A). In particular, double mutant clone 1 has less of an impact. Is there thought to be suppressor? What underlies these differences?
10. The TRFs in Fig. %C of the *trf2^{crg1}* strains grown on YPD show bulk signal ranging below 0.25 to just below 0.5 kb, whereas the analysis of similar strains in Fig. 5E show an abundance of signal in the range above 0.5 and below 0.75 kb. Why the difference?
11. A direct comparison of C-circles in the *ku70^{nar1}* and *trf2^{crg1}* strains is needed before any conclusions can be drawn regarding whether differences exist or not (end of page 8). For example, can double mutant conditional strains be generated and studied?
12. Sequencing of products detected in the telomere-telomere fusion assay and demonstration of the requirement of both primers is needed to validate the assay. Minor issues
 1. The abstract points to an “unprecedented division of labor”. That seems like an overstatement given the distinct roles of mammalian Trf1 and Trf2 proteins, which are noted well in the introduction.
 2. Fig. 1A The green color of the insertion blocks of the Trf2 and Trf2ΔN make it somewhat difficult to see the purple outline of the Myb domain.
 3. Fig. 1B Are the Trf1 proteins not C-terminally tagged with FG as in 1C?
 4. Fig. 1B Figure legend should indicate the various nM values or the specific fold series (e.g., it seems like it is a 3X-fold series; is it?).
 5. Fig. 1B The quantification suggests that Trf1 might have more than “slightly” higher affinity for the (TTAGGG)₄ DNA than Trf1^{N274}. Were other results quantified? Can the results from replicates be averaged and standard deviations be determined?
 6. (TTAGGG)₄ and TTAGGG₄ are variably used. Use of one or the other would be best.
 7. Fig. 1C – FG needs to be spelled out initially and described.
 8. Fig. 1C – Cg TEL is not a G-rich repeat; it is G-rich but not a repeat.

9. The methods for Figure 3C are not described.
10. Pg. 8, lane 3. In the absence of statistical analyses, the term “significantly” should be removed.

Reviewers' comments:

Reviewer #1 (Remarks to the Author):

The paper by Yu et al. provides evidence that *Ustilago maydis* employs two distinct telomere-binding proteins (TBPs) that play complementary functions in telomere maintenance in this organism. While UmTrf1 seems to play a primary role in telomere replication, UmTrf2 is involved in telomere protection.

The results of the study are very interesting in terms of both (1) mechanism of telomere maintenance in *U. maydis*; and (2) evolution of telomeres. The experiments are very well designed and executed and represent a complex mix of in vitro and in vivo assays resulting in a nice, highly entertaining story.

We thank the reviewer for the comments and agree that the mix of in vitro and in vivo assays is a strength of the paper.

I believe that the paper is an important addition to the field and I have only a few comments:

1. p.3: the sentence „TRF1, on the other hand, ...“ indicates that only TRF1 recruits BLM helicase to telomeres. As shown by Lillard-Wetherell et al. (Hum. Mol. Genet., 2004) and as mentioned by the authors in the other parts of the manuscript, both TRF1 and TRF2 associate with Blm.

The in vivo function of TRF1 in recruiting BLM was confirmed by mutations that disrupt their mutual interaction (Zimmermann et al., Genes Dev. 28, 2477). In contrast, while Lillard-Wetherell et al. described an interaction between TRF2 and BLM, the in vivo significance of this remains unclear. We have added a sentence on this finding (page 3, middle).

2. p.3: first mention of *Schizosaccharomyces pombe* should be in unabbreviated form
Corrected (page 3, middle).

3. p.3: sentence „The other protein...“ the reference to Fig. 1 does not seem appropriate. The reference to Fig. 1 is eliminated.

4. p.4: SpTay1 is not an official name of the protein, I suggest to call it SpTeb1 as indicated in PomBase database: <https://www.pombase.org/gene/SPAC13G7.10>
Corrected (page 3).

5. The terminology of the various types of TBPs is somewhat confusing. UmTrf1 (and SpTeb1) are Tay1-like (this is clear), UmTrf2 is described as TRF and Tbf1-like. Is it really true that yeast Tbf1 proteins (such as ScTbf1, CaTbf1 and SpTbf1) are homologues of human TRF1/TRF2, or do they represent another group of TBPs? Do these proteins contain TRFH domain homologous to mammalian TRFs? Even Taz1 of *S. pombe*, originally thought to be a homologue of TRF1/TRF2 was shown to be structurally unrelated protein (<https://www.ncbi.nlm.nih.gov/pmc/articles/PMC4493283/>) so calling UmTrf2 as TRF1/TRF2-like is a little far-fetched.

As shown in Supp. Fig. 1A, an N-terminal region of UmTrf2 aligns well to the TRFH domains of vertebrate TRF proteins. This is also true of ScTbf1 and SpTbf1 (see the alignment at the end of this file). We have now indicated that the designation of "TRF and Tbf1-like" is based on sequence alignment (page 4).

6. The discussion implies that Tbf1 protein was lost in *Yarrowia lipolytica*. However, this is not the case. As originally described by Kramara et al. (2010), there is a gene YALI0B18018 encoding a homologue of ScTbf1, whose function at telomeres of *Y. lipolytica* was so far not carefully assessed. Thus the text as well as a scheme presented on Figure 7 should be corrected.

We have modified Fig. 7 and related text to include the discussion of YITbf1 (page 11).

7. p.5 and Figure 1 legend: CgTEL oligo: it should be indicated that it is derived from the telomeric repeat of *Candida glabrata*; YITEL --- from *Yarrowia lipolytica*

Modified as recommended (page 5, first paragraph).

8. p.5: The sentence: „In addition...“: a reference to Fig. 1A would be more appropriate. **Corrected.**

9. TRF analyses presented on Figures 3-5: In some cases, the differences in lengths of TRFs do not seem as obvious as claimed by the authors. Therefore, the lengths of TRFs should be subjected to quantitative analysis. In addition, the intensity of the signals varies between the lanes. As no loading control(s) are presented, it is not clear, if this variation is due to different amounts of total DNA or different relative amount of telomeric sequences.

We have quantified average telomere lengths for Fig. 3A and Fig. 4C and plotted the results in the respective figures. For Fig. 5, the dramatic increase in telomere lengths heterogeneity in the mutant is visually clear and does not require quantitation to reach this conclusion. The intensity variation from lane to lane in Fig. 5E was due to different amounts of total DNA as judged from EtBr staining of the same gel.

0. Figure 3C: the mutant *trf1Δ* seems to exhibit a different growth on the three upper plates. Is it reproducible? Yes, clone 1 appears to grow consistently faster than clone 2. We do not know the reason behind this; it could be due to some unrelated mutations introduced during gene replacement. Both strains show greater sensitivity to MMS, but not to other clastogens, indicating that this is a consistent property of *trf1Δ*.

1. Figure 4: Is there a reason why the authors used different concentrations of both Blm and Trf1 in experiments presented as Fig. 4A and 4B respectively. In legend to Figure 4D it should be indicated, what represent the numbers 1-3 above the lanes (independent clones? Different passages?)

The different concentrations of Blm used for Fig. 4A and 4B were chosen to accentuate the stimulatory and inhibitor effects of Trf1 on different substrates. This is now explained in the figure legend (page 21). The *ku70 nar1 trf1Δ* samples 1-3 in Figure 4D are from three independently constructed mutant strains. This is now explained in figure legend (page 21).

2. Figure 5B should contain a scale bar.

A scale bar has been added.

3. Although it was described by de Sena-Tomas et al. (2015) the authors should briefly (e.g. in Methods section) explain what is the basis regulation of *ku70nar1* allele.

This information is now included (page 13).

14. The authors claim that division of labor between UmTrf1 and UmTrf2 are unprecedented, but mammalian TRF1 and TRF2 (even though they, in contrast to UmTrf1 and UmTrf2 are homologues) also exhibit division of labor at telomeres.

This characterization is removed from the text.

15. In the discussion, the affinities of UmTrf1 and UmTrf2 to TTAGGG and TTAGGG-like repeats should be compared with KD obtained for other Tay1-like and Tbf1-like proteins, respectively, in previously published studies.

Because different studies employ different oligonucleotide substrates, the comparison across studies is not particularly informative. We mention some earlier results in the Fig. 1B legend and discuss the limitation of comparing across different studies (page 20).

16. p. 10: the sentence „The Tay1 protein family...“: there should be a reference to Figure 7 (not Figure 6). Corrected.

17. The discussion on evolution of yeast telomere-binding proteins (visualized on Figure 7) could be extended also to evolution of Rap1 and Taz1 as a major TBPs in some phylogenetic branches and compare their current views with the scenarios previously proposed by the authors' and other laboratories.

The evolution of Rap1 and Taz1 as the major TBP in budding and fission yeasts are certainly quite interesting. However, given that the current study does not bear directly on these protein families, discussing them at length may detract from the main focus of the paper. We have alluded to Rap1 and Taz1 in passing and provided two references for interested readers (page 12, near the top).

Reviewer #2 (Remarks to the Author):

The experiments reported in this manuscript describe telomere maintenance functions of two fungal Myb-containing proteins that have been found based on sequence analogy with human TRF1 and TRF2 Myb domains. The experimentation is performed mainly in vitro with purified bacteria-expressed variants of proteins: UmTrf1 was expressed in full length but UmTrf2 was expressed without N-terminal 415 AA residues.

This appears to be the second report that examines how fungal proteins containing Myb domain interact with telomeric DNA. As far as I know, the first study describing telomeric proteins in fungi has been published by the Tomaska laboratory (Kramara J, et al. Tay1 protein, a novel telomere binding factor from *Yarrowia lipolytica*. J Biol Chem, 2010).

The key findings of recent study are that UmTrf1 is required for telomere maintenance, UmTrf1 modulates Blm helicase and UmTrf2 is essential for cell survival and sustainable telomere protection. I am afraid that the conclusions drawn are insufficiently supported by the data. The measured data and conclusions could have been more robust if additional data points and replicates are measured and presented (see below). Given the level of detail provided, methodology information and validity of statistical analyses are insufficient, especially in regards to the ability of a researcher to reproduce the work. Taken together, the major conceptual drawback for me is the fact that the authors performed most of the experiments with truncated variant of UmTrf2 lacking N-terminal part. In case of human TRF2, N-terminal part is responsible for additional DNA binding modes. Thus, if only the truncated variant is investigated, we can miss important functional contributions of N-terminal part of UmTrf2. The manuscript contains good initial data, but in present form, it needs more carefully conducted experiments with full-length proteins to the extent that significantly diminishes my enthusiasm for the paper. Therefore, I suggest rejecting the manuscript in present form.

The authors could consider kindly suggestions that might improve their experimental concept and

manuscript. 1. Full-length UmTrf2 should be expressed using a yeast expression system.

We agree that it would be ideal to analyze the DNA-binding property of the full length protein. However, we have not been successful in this. It is worth noting that the region deleted from Trf2 Δ N-FG (the first 414 amino acids) is absent from metazoans and poorly conserved among fungal species (see the alignment at the end of this file), suggesting that it is not critical for function. In addition, since the main goal of the DNA-binding study is to show that Trf2 has adequate affinity and specificity for telomere DNA, the positive results we obtained with Trf2 Δ N-FG are sufficient for this conclusion. We have added a few sentences in the text to explain this (top of page 6).

2. EMSA assays should be carried out with at least nine data points and in three replicates to obtain reliable and robust conclusions.

Our goal was to confirm that Trf1 and Trf2 manifest K_d s and DNA binding specificities that are consistent with direct telomere functions in vivo. We have now performed all EMSA assays with 4-5 data points each and in triplicates (Fig. 1B, 1C, 2B, 2C). We believe that results in total provide strong support for our major conclusions.

3. Statistical analyses and fitting functions should be described clearly. It is not obvious how the authors determined K_d values.

We performed the binding assays in triplicates and fitted the averages of the data points to a curve for one site binding (hyperbola) model in GraphPad Prism (GraphPad Software Inc.). This is now described in Methods (top of page 15).

4. Methods should be provided with all details to the level that enables a researcher to reproduce the work.

For example:

a. "C-circle assays were carried out by the rolling circle amplification method (22), with modifications that were also previously described (19)." should be described precisely.

We have included details of this and other assays (page 14).

b. Electrophoresis assays should be provided with electric field E values in V/cm.

Provided (page 14 and 15).

Reviewer #3 (Remarks to the Author):

In this manuscript, Yu et al., characterize two distinct duplex telomere repeat binding proteins in the fungus *Ustilago maydis*, which shares the mammalian telomeric repeat, in contrast to other studied fungi in the ascomycetes phyla. They present findings related to the ability of these proteins, *UmTrf1* and *UmTrf2*, which are referred to hereafter as simply Trf1 and Trf2, to bind various telomeric repeat containing substrates despite the presence of multiple short insertions within the TRFH- and Myb-like domains. They additionally present data suggesting Trf1 is required for maintenance of long telomere tracts and stimulates ALT-like telomere recombination, activities that may be achieved through modulating the activity of Blm. They further demonstrate that, in contrast to Trf1, Trf2 is essential and is involved in telomere end protection (inhibition of ECTR formation and C-strand SS DNA). The finding that Trf1, like Blm, suppresses the ALT-like telomere phenotype in *ku70nar1* cells is quite striking, as is the impact of conditional suppression of Trf2 on telomere length and integrity. Lastly, the development of an assay to detect telomere-telomere fusions in *U. maydis* is a potential technical strength (see below). The authors use these findings to ponder in the discussion the plasticity of the Tay1 protein family, to which Trf1 belongs. While the findings and discussion are interesting, there are several experimental details that need to be addressed:

Major issues

1. There appears to be some variability of the impact of *trf1 Δ* on telomere length. In Fig. 3A, the bulk TRF lengths in clone #3 appear similar to WT. The lengths in clone 1 seem uniformly decreased rather than exhibiting just shortening of the longer telomeres. Moreover, the STELA analysis is unclear. Typically, multiple independent reactions are prepared from DNA isolated from a single culture. I assume this is the case here as well. Thus, while the STELA analysis of the single WT and *trf1 Δ* clones looks convincing for a possible selective effect of loss of Trf1 on long telomeres, an analysis of multiple clones is needed.

We have included STELA analysis of multiple *trf1 Δ* clones in Fig. 3B.

2. Data demonstrating telomerase-dependent or -independent shortening or defects in replication of duplex telomeric repeats is needed to conclude that Trf1 contributes to telomere replication.

This turned out to be a challenging experiment. Because telomere loss in the telomerase-deficient mutant is progressive and passage-dependent, the comparison of single and double mutants should ideally be performed using cells in which the genes were knocked-out or repressed at the same time. Accordingly, we placed *trt1* (TERT), *trf1*, or both genes under the control of the arabinose-dependent *crg1* promoter and used YP-arabinose and YP-dextrose to regulate the expression of either or both genes simultaneously. As shown in Supplementary Fig. 4, we found that at the equivalent passage following transcriptional repression, the *tert_{crg1} trf1^{crg1}* double mutant manifests accelerated telomere loss relative to the single mutants, supporting the idea that these two genes function in different pathways of telomere maintenance (bottom of page 6).

3. The description of the telomeric probes in Supp. Fig 2 might be incorrect. For example, the sequence of TTAGGG₄ is different than in Fig. 1C and has a repeat of GGGTAA. The sequences of the other probes are also in question. We have corrected the mistakes.

4. The interpretation of the Supp. Fig. 2 needs reconsideration. First, what is the basis for the templates designating TTAGGG₂ and TTAGGG₄ having two and four repeats, respectively, and that there is any functional relevance to a half repeat or a repeat split across the ends of a probe? An alternative interpretation of the data in Supp. Fig. 2 is that Trf1-TG has equivalent binding to the TTAGGG₄ and TTAGGG_{3.5} probes, which each has 3 complete repeats; binds less to the TTAGGG_{2.5} probe which has just 2 complete repeats; and does not bind the TTAGGG₂ probe which has just a single repeat.

The reviewer's point is well taken. We have now adopted this interpretation, which is more consistent with crystallographic data that highlight the significance of GGGTTA as the binding target of individual Myb motif (page 6, line 8-11).

5. The growth properties of the *trf1Δ* clones in Fig. 3C appears strikingly different than the growth of the WT and *rad51Δ*. This is particularly true on YEPS and DEB and after UV treatment. Moreover, MMS has a rather dramatic effect on this property, suggesting the strains have more than a slight sensitivity.

We have modified the discussion of Fig. 3C (page 7, near the top).

6. Fig. 4B and Supp. Fig.4B How are the fractions unwound calculated given the two distinct bands in the absence of protein added? How is it that the telomeric DNA substrates form a stable dimeric structure if the overhangs are not complementary? Why would the "dimeric" molecule be preferentially unwound by BLM (compare e.g., top and middle bands in 4B lane 1 with comparable bands in lane 2; same type of analysis could be done for Supp. Fig. 4)? Lastly, how reproducible are the effects observed in these helicase activity assays (quantification appears to be done on single experiments).

To determine the fraction unwound by Blm, we calculated the ratio of labeled DNA in ssDNA form (the bottom species in the gel) relative to total labeled DNA signal (described in Method). The telomeric DNA substrate contains three copies of GGG triplet in the 3' ssDNA region, which is sufficient to engage in non-standard, Hoogsteen base interactions such as in G4. As the reviewer pointed out, our data suggest that Blm may work preferentially on the dimeric substrate. This is not surprising given the ample literature on the ability of Blm to unwind G4 DNA (e.g., Sun et al., JBC 273, 27587) (page 7, middle). We have not investigated this further since this will require substantially more quantitative analysis, and is beyond the scope of the current study. The stimulatory and inhibitory effects of Trf1 on Blm helicase were observed in at least three independent experiments. We have not combined the quantitation from multiple experiments because these experiments used different substrate and protein concentrations.

7. What is the most rapidly migrating band in the telomeric DNA lanes in Suppl. Fig 3? The putative single stranded labeled molecule is designated to the right. Does that make sense?

This is now Supp. Fig. 5. We have corrected the designation for the labeled ssDNA, which corresponds to the most rapidly migrating species in the gel.

8. The median and 3rd interquartile values for *blmΔ* and *trf1Δ blm1Δ* do not appear different in Suppl. Fig. 5B, so these data do not support the conclusion of a more severe effect in the double mutant. Additionally, there appears to be a low of the shorter fragments in the STELA experiment but not in the TRF analysis. Could it be that the shorter fragments detected by Southern are not amplified in the STELA?
The difference in telomeres between *blmΔ* and *trf1Δ blm1Δ* is most obvious in the upper range of the telomere length distribution. The longest STELA fragments in *blmΔ* (~1.3 to 1.5 kb) are mostly absent in *trf1Δ blm1Δ*, resulting in a significant decrease in the first quartile boundary. In contrast, the shorter telomeres in the two mutants do not exhibit as much difference. We have described this in more detail in the revision (page 7, near the bottom).
9. The TRF, c-circle, and C strand ssDNA analyses comparing the *ku70nar1* and *trf1Δ ku70nar1* strains shows clear differences, however, the suppression of the telomere length heterogeneity and c-circles is quite variable (Figs. 4D, 4E, Suppl. Fig 7A). In particular, double mutant clone 1 has less of an impact. Is there thought to be suppressor? What underlies these differences?
Yes, double mutant clone 1 exhibited less suppression of the various telomere defects. We do not know the underlying reason, but the existence an extra mutation is possible. We have noted this in the paper (page 8, middle).
10. The TRFs in Fig. 5C of the *trf2crg1* strains grown on YPD show bulk signal ranging below 0.25 to just below 0.5 kb, whereas the analysis of similar strains in Fig. 5E show an abundance of signal in the range above 0.5 and below 0.75 kb. Why the difference?
The Fig. 5E gel has uneven migration of samples near the edge of the gel such that the standards migrated further than the majority of samples, resulting in biased estimates of telomere size. We have re-assessed the correct positions of the size standards based on the curvature of loading dyes, and the new estimates showed that the peaks of low molecular weight signals are similar in Fig. 5C and 5E.
11. A direct comparison of C-circles in the *ku70nar1* and *trf2crg1* strains is needed before any conclusions can be drawn regarding whether differences exist or not (end of page 8). For example, can double mutant conditional strains be generated and studied?
The two sets of C-circle assays were performed concurrently and analyzed on the same blot. We had previously compressed the scale for the *trf2^{crg1}* portion of the blot and normalized the signals to the *trf2^{crg1}/YPD* samples for Fig. 6B. To make it easier to compare between the *ku70^{nar1}* and *trf2^{crg1}* assays, we now show the blots and the plots using the same scale (Fig. 4E and Fig. 6B) (page 9, second paragraph). The data show clear differences in the levels of C-circles. Whether this represents a mechanistic difference or a difference in degree is not clear.
12. Sequencing of products detected in the telomere-telomere fusion assay and demonstration of the requirement of both primers is needed to validate the assay.
We have shown that the vast majority of products from the fusion PCR reactions hybridized to both UT4/5 and UT6 subtelomere probes (Supp. Fig. 10A). We have also confirmed by sequencing a cloned product that the two ends of this PCR product correspond to the two types of subtelomeres (Supp. Fig. 10B). These results support the validity of the fusion assay (page 9, third paragraph).

Minor issues

1. The abstract points to an “unprecedented division of labor”. That seems like an overstatement given the distinct roles of mammalian Trf1 and Trf2 proteins, which are noted well in the introduction.
This characterization is removed.
2. Fig. 1A The green color of the insertion blocks of the Trf2 and Trf2ΔN make it somewhat difficult to see the purple outline of the Myb domain.
The figure is modified.

3. Fig. 1B Are the Trf1 proteins not C-terminally tagged with FG as in 1C?
This is now explained in the figure legend (page 20). Because we also indicate in the figure the sizes of the proteins, adding the tag to the illustration may introduce confusions.
4. Fig. 1B Figure legend should indicate the various nM values or the specific fold series (e.g., it seems like it is a 3X-fold series; is it?).
Yes, indicated.
5. Fig. 1B. The quantification suggests that Trf1 might have more than “slightly” higher affinity for the (TTAGGG)₄ DNA than Trf1N274. Were other results quantified? Can the results from replicates be averaged and standard deviations be determined?
The plot in Fig. 1B now incorporates data from three independent series of assays, and the estimated K_ds for TRF1 and TRF1^{N274} are stated in the text (page 5, first paragraph).
6. (TTAGGG)₄ and TTAGGG₄ are variably used. Use of one or the other would be best. We now use TTAGGG₄ throughout.
7. Fig. 1C – FG needs to be spelled out initially and described.
This is now explained in Fig. 1B legend.
8. Fig. 1C – Cg TEL is not a G-rich repeat; it is G-rich but not a repeat.
We now refer to the CgTEL oligo or sequence (page 5).
9. The methods for Figure 3C are not described.
Described (page 21).
10. Pg. 8, line 3. In the absence of statistical analyses, the term “significantly” should be removed. Removed.

Alignment of yeast Tbf1 and vertebrate TRF1

TRFH domain

```

Scer      11  -----S-----NRFNDIIQSLPARR
Cgla     14  -----D-----VNRFNANIEDLPEKTR
Scas      9  -----N-----NRFNNIIQALELPRTR
Klac      9  -----D-----HVFDSILRRLEPEKTQ
Agos     16  -----L-----KQRLAVLPLGLPYATQ
Cgui    213  TN TAF AAYTA SSSLPPL-----ASV TSAH AALPLPI APDYVAPRIQ
Calb    267  TN QAY AAYTS SSSLEQHTS--ASAMLSSATLSALPLSI APVYLPPRIQ
Dhan    304  TN TAY AAYNA SSSQLPPL-----SILANAH AALPLPI AADYLPPRIQ
Cpar    206  TN AAHVAYSS SSSQYPHLNVD SRHAVFTQAN SVLPLPI GADFLPPRIQ
Sjap     47  PDVSAA-GLN NFDLPGIPLNMNDFYMRVNGMDFGFSQNSIIAHPQHV
Soct     46  PNVASNGDIP LGFDLSGIPINMTPDFYLRMNQMEYAFNQPNPVASPOYL
Spom     47  QNVAPNAEIP LGFDLSGIQFNMTPDFYLRMNQGM DYAFNQPNPIATPQQL
Hsap     47  COVQVGAPEEEEEEEEDAGLVAEAEVAAGWMLDFLCLSCRAFRDGRSE
Mmus     38  COLQLGTPREMEAE----LVAEVEVAAGWMLDFLCLSCRAFRDGRSE
Xlea     10  -----FDDTAAVATNWCDFMFASCFYFREDRSE
Ggal     18  -----SALAEVAADWVLEFSCCCCRYFVVECEA
conserve 401  . . . . .

```

```

Scer     28  I T I C S L C L L N I S T Q L L R F I L N A N S P N I A V L T D Q T A F -- L S S C E T E I F
Cgla    31  I T L S T L S L N D V S T Q L L R F I S N A H T P N V I A V I S N K E N Y -- I S T C E S E M F
Scas    26  I T I S S L C L L N I S T Q L L R F I F N A N S P H A T A I L T D N K A S P Y S Q C E T D T F
Klac    26  I L L K T L P L L G V T V Q L L R F S A N S L R - T I A I S D A S P K -- A F P L E Q D L Y
Agos    33  I T L K S L P L N D I S T Q L L R F I T I P L H V L -- T G S V - Q E D K -- A A Q A E H E L F
Cgui   257  I L V N S L P L N L A T Q I L R M V A V G P Y Q - K V V E L A S N P D -----S A A G A A Y
Calb   315  I L I N T L P L N L A T Q L L R I V A T S P Y Q - K I D I A S N P D -----T S A G A T Y
Dhan   348  I L I N T L P L N L A S Q L L R I V A L G P Y Q - K I D I A A H P D -----T P A G A T F
Cpar   256  I L I N T L P L N L S T Q L L R I V A K S S Y K - K I D I A S N A E -----T P E G A T Y
Sjap    96  I R T S L I P I L G N L S A V I L S I L G K - P F E - E A S I V T N P D -----S E M G I A F
Soct    96  I R S S L V P I L A N L S N I I L S I L G K - P V Q - E A S I V T N P A -----S E M G M A F
Spom    97  I R T S L I P I L G N L S N I I L S I L G K - P V Q - E A S I V T N P A -----S E M G M A F
Hsap    97  D F R R T R N S A E A I I H G L S S T A C Q L R T I Y I C Q F L T R I A -----A G -----
Mmus    84  D F R R T R D S A E A I I H G L H R T A Y Q L K T V Y I C Q F L T R V A -----S G -----
Xlea    40  D F Q R S T H M L E W L L E G S Q K I D A H R K T I - P I A Q F L M R V A -----E G -----
Ggal    48  E F R R W R D V A H A V S N G F S K V T H T H Q K M V Y I C Q L L I R I A -----E G -----
conserve 451  . . . . .

```

```

Scer     76  Q T I V K I E K Q I R M I Y H T R S P L L S V H D V A P G I W F P N S P P P L I L R G H E A F I I T
Cgla     79  H T I V R I E K N I R D V Y E S R P L L N V N N V A P G I W L L N M P P P L I L K G H E A Y I I S
Scas     76  Q T I L K I E K Q V R M I Y N S R Y P L L T V H D V A P G I W F P D S A P P S L L R G H E A Y I I T
Klac     73  H S L I A L E I K V S L Y N T K P Q L L R V S H I F P G V W H D H H P S P S F I Q T H E R Y I L L
Agos     78  H V I V D A L I E V K E V Y G D - I S L L T V Y D V A P G I W F P G C D P P L I L R N F E Q F L A
Cgui   300  R D L T S L F E F T K R I Y S E E D P P L S V E H I A P G M W K E G D Q T P N F K N R E Q S I E S
Calb   358  R D L T S L F E F T K R I Y S E E D P P L I V E H I A P G M W K E G E E T P S F E K P K O O S I E S
Dhan   391  R D L T S L F E F T K R I Y S E E D P P L S V E H I A P G M W K E G D K T P S F E N R E Q S I E S
Cpar   299  R D L T S L F E F T K R I Y S E E D P P L I V D H I A P G M W H T G E E T P R F E K N K E Q S I E S
Sjap    138  A K V M M E R M K E M Y S E D S F - I -----Y P D A I N M K T N A O K T
Soct    138  T K I M S M E R M K D I Y T E D S F - I -----Y P A V I S M R T P S Q R S
Spom    139  T K V M M E R M K D I Y T E E S F - I -----Y S S A I G M R T P S Q R S
Hsap    136  K T I D A Q F E V D E R I T P L E S A L M I W -----G S E K E H D K ---
Mmus    123  K A I D A Q F E V D E R I T P L E S A L M I W -----N S E K E H D K ---
Xlea     78  K N I D S Q F T D E S I T P L E A L M A F -----N Q I - E E E E D K H
Ggal     87  K R L E C H F E N N T T I S P L E S A L S F W -----T L L E R E S K N T
conserve 501  . . . . .

```

Scer 126 ATRKANLTFLLTSLNCLNYGRELLOSIFLDI-FCPNTNTVGNNS-----
 Cgla 129 TIRKANLTFLLTVINCIPYGEDFLOKVLDV-FCPNTILTDQTN-----
 Scas 126 ATRKVNLTFTLTALGSFHYGRELLOSTFLDI-FCPNTLYNSSNSTT---
 Klac 123 VERKVNICFLLTLKELPFCISFLADNFLEV-FCFNVDSKVEDLITA-
 Agos 127 TIRKCNLVMFLLTLGKFQYGEQFLNECFLDI-FCPSNSYSTSDGGKDT-
 Cgui 350 TIRKVNLTFTVAALGTMEVGFYYLNSEFFLN-FCPPHGLDVENSMSNM-
 Calb 408 TIRKVNLTFTLAATLGTMEVGFYYLNSEFLDV-FCPSNNLDPSNALS-
 Dhan 441 TIRKVNLTFTGATLGTMEVGFYYLNSEFLDI-FCPANLDPNTTLSNM-
 Cpar 349 TIRKVNLTFTAAALGTMEVGFYYLNSEFLDI-FCPAHDLDIKYHLSDSS-
 Sjap 172 ATRRANLAFLLAAVYCALQVGFHHLNENFLEV-FAPD-----
 Soct 172 AVRRANLAFLLSAVYCALQVGFHHLNENFLEV-FAPD-----
 Spom 173 TTRRANLAFLLAAVYCALQVGFHHLNENFLEV-FAPD-----
 Hsap 168 LHEETONLTKQAVAVCMENGNFKEAEVEVERIEFGDPNSHMP-----
 Mmus 155 LHDEIKNLTKQAVAVCMENGSFKEAEVEVERIEFGDPEFYTP-----
 Xlea 112 LHEETEELLTKQAVVTCMEKCRKLSAETLDRLEKESGSKNY-----
 Ggal 122 LHEETRRLIQVAVVYMEKCYKEAAEVLRLLETDSSESHKP-----
 conserve 551*.....*..

Scer 170 -----L-----E-----QSKFKLKSQAIIYLDLKTQA
 Cgla 173 -----G-----D-----QVSKFKLKPQAIIYLDLKTQA
 Scas 172 -----MG-----E-----PNCKFKLKSQAVLYLDLKTQA
 Klac 171 -----NR-----S-----RIVYGRLLKQPQVLYLDLMTQY
 Agos 175 -----GSYSVQSNASFTPCSTSTFSGLTSAIGRLLKHTHTLYLNMKTQA
 Cgui 398 -----SAEN-----MNLQTGANGDAGRLLKPOAQLEFLDLKTQA
 Calb 455 -----LGGYQNG-----LQS-TDSPVGARVVKLLKPOATLYLDLKTQA
 Dhan 489 -----SPHNMS-----LQSGVNTIIGDKVVKLLKQPQVLYLDLKTQA
 Cpar 398 YLDTANSLATT-----IQSGAGSNVGGKLVKLLKPOAMLYLDLKTQA
 Sjap 208 -----GCKILANQGVLFEMELKTQA
 Soct 208 -----EANTILTNQCTLYMELKTQA
 Spom 209 -----ESNIIITNQCFLYMELKTQA
 Hsap 210 -----FKSKLIMTISQKDTFHSFFQHFS
 Mmus 197 -----LERKLLKIMISQKDVFHSLFQHFS
 Xlea 154 -----LRMKLTMLEKQDPYHEFLQNFT
 Ggal 164 -----LRMKLAIVKSKDPYVPLIQSFS
 conserve 601

Scer 192 YIAGLKEFQDETNEIISLEKKQELLDLFFPSNADILVQRRTGDSGD----
 Cgla 195 YISSIRDHLGANEQIPDAEKRIILDTHFPNDAMRLVNRIVAPNE----
 Scas 195 EISALKDYEDDSNVIPQGGKQELLNLVFPANPHSLIKRRFGAPEDTITN
 Klac 196 YISALVSSS-----TPHTSKSPLLDQVFPITLDDIKDTIKYAGKYMQ-
 Agos 220 FIAAMEQED-----PLLNRQIILDTHFPADMATYLOLSGEPAN--E-
 Cgui 431 YISALEVGRSREDI-----LEDILPDDIEPVLLAWNSRM-----
 Calb 492 YISALEAGRSKEEI-----LEDILPDDIEHVYLSRRNAKI-----
 Dhan 526 YISALEAGGRSREDI-----LEDILPHNDEIDILERRKTKI-----
 Cpar 440 YISALEAGRSREEI-----LEDILPNDQIILLARNVDL-----
 Sjap 227 YISAMAQARSKEEI-----LNDLFPDPAQQFLRRRASIT-----
 Soct 227 YISAMAQARPKEDI-----LNDLFPSDMAHRFLRRNAKLD-----
 Spom 228 YISAMAQARPKEDI-----LNDLFPSDMAHRFLIRNAKLD-----
 Hsap 233 YNHMMEKIK---SY-----VNYVLSKSSSTFLKAAAKVVEK--
 Mmus 220 YSCMMEKIQ---SY-----VGDVLSKSSSTFLKAAATKVVEN-E--
 Xlea 177 YAQMCKKIK---SY-----IALKMKRPSVFLKAAAKVVEA-TA-
 Ggal 187 YSLLSKVK---SY-----VKLFLKRNRTNFIQAATKQVES-E--
 conserve 651

Myb/Sant domain

Scer 382 SSGP-----HSS-H-NNSS--NSNNN-G-SIGLRK---PKAKRTWSKE
 Cgla 395 HMSGGENVHSQLVD-S-NNSI--LSDVI-SKSPSVKV---VKAKRTWSKE
 Scas 411 SIDSRLQOSVIHAA-VANSAI---SSNN-VSGNVHKK---LKPKRVWSKE
 Klac 346 EAEVDVKSTVLRT-----NEPQRIDL-PVKPFKTK---PKQKRMWTOE
 Agos 375 TDGVKAPNGVSSC-----SVP-SGHLN-EPIIARKK---LQOKRMWVKE
 Cgui 586 -----SE-IQ-----EQQIHLRFNPNSTVR---NEORRPWTRD
 Calb 637 -----SE-MR-----EANIAPV-----K---PSORRAWSRE
 Dhan 682 -----SE-IQ-----EQQIHLRLNPGSNVK---SLORRPWTRD
 Cpar 583 -----SE-IR-----EQQIHVNGH-----K---PSORRPWTRD
 Sjap 382 -----EQ-VR-----RMVG-----SSAPRK---PTNRRSWTKE
 Soct 383 -----DQ-VR-----SMTNTG-NGNKRTKK---VANRRRTWTKE
 Spom 384 -----DQ-VR-----HMTNNN-LNNKRTRR---VANRRRSWTKE
 Hsap 361 -----RIPVS-----KSQPVTP EK-----HRAFKRQAWLWE
 Mmus 348 -----RLCVS-----ENQPD TDDK-----SGRFKRQATLWE
 Xlea 336 -----GK-NS-----KENQENVKD-SRTEKPLNSK KROHWTWE
 Ggal 299 -----C-----SERRROPWTYE
 conserve 901 *

Scer 416 EEEALVEGLKEV-GPSWSKILLYGPGGKLTENLKNRTQVQLKDKARNWK
 Cgla 437 EEKTLITGLIDL-GPAWAKILLYGPGGKLTENLKNRTQVQLKDKARNWK
 Scas 453 EEATLLEGLQEV-GPSWSKILLYGPGGKLTENLKNRTQVQLKDKARNWK
 Klac 386 EEDCLKSGLRQC-GPAWAKILLSYGPGGTVSESLKNRSQVQLKDKARNWK
 Agos 414 EEEALISALKVY-GPAWSKILLYHCGAGGSVSETLKNRTQVQLKDKARNWK
 Cgui 615 EEKALRQALELK-GPOWSTILELEFGAGGKISEALKNRTQVQLKDKARNWK
 Calb 659 EEKALRHALELK-GPHWATILELEFGGGKISEALKNRTQVQLKDKARNWK
 Dhan 711 EEKALRHALELK-GPSWSNILELEFGAGGKISEALKNRSQVQLKDKARNWK
 Cpar 607 EEKAFRHALELN-GPHWSKILELEFGGGKISEALKNRTQVQLKDKARNWK
 Sjap 406 EEEALLEGLDQVKGPKWSQILELYGPGGKSEVLKDRNQVQLKDKARNWK
 Soct 411 EEEALLDGLDLVKGPRWSQILELYGPGGKNEVILKHRNQVQLKDKARNWK
 Spom 412 EEEALLDGLDLVKGPRWSQILELYGPGGKSEVLKYRNOVQLKDKARNWK
 Hsap 387 EDKNLRSGVRKYEGGNWSKILLHY-----KFNNRTSVMLKDRWRITMK
 Mmus 374 EDRIILKCGVRKYEGGNWAKILLSHY-----KFNNRTSVMLKDRWRITMK
 Xlea 367 EDLILKCGVRKFGVGNWSKILLHY-----EFNNRTGVMLKDRWRITMK
 Ggal 310 EDKIKLKSQVREFGVGNWIKILLHG-----DFNNRTSVMLKDRWRITLC
 conserve 951 * * * * *

Scer 465 IQYLKSGKPLEDYLIKVTGNLEKIYKAKKFSQSPNSSTIMEQNLSQH-P
 Cgla 486 IHMLKTKKPLPEYLNRVTNLDKIAKTRRGS-----MQQQVQSQH-D
 Scas 502 IQYLKNNKPLEDYLEKVTGNLDKVKSKKKARI-----MQQQVQSQH-D
 Klac 435 MHMLKNNKPLPEYLEKVTGDLERGLKSKKK-----PKSK-----
 Agos 463 MHMLKKNKPLPEYLLKVTGNLREEKFKRKSAKRKS KSGSPS-----
 Cgui 664 MFELKTKMPVETYLQKVTGDLEREDRSREAKR-SRSRKTAAAPVPSI---
 Calb 708 KFELRSGLEIPSYLRGVTGGVDD--GKR--KKNVTKKTAAPVPMNS-E
 Dhan 760 MFELKSGLEIPAYLQKVTGDLERDDRSQNKRLNRRKTAAPVPTPIQN
 Cpar 656 VFYLLKNGEMIPSYLKTVTGNLED--RVK--RSVSRQEKTAAPVNVH-N
 Sjap 456 LFELKNGQMVEPFLQFVTGDLERD-----
 Soct 461 LFELKSGQIVEAPLQCVTGDLRRE-----
 Spom 462 LFELKSGQIVEAPLQCVTGDLRD-----
 Hsap 429 KL-----KLISDSED-----
 Mmus 416 RL-----KLIS-----
 Xlea 409 RL-----KIVDSICDL-----
 Ggal 352 KI-----K-----
 conserve 1001

Alignment UmTrf2 N-terminus

fungi

```

Hs_TRF1      1  ---MAEDVSSA-----APSP-----RG---CA-----
Mm_TRF1      1  MAETVSSAARD-----APSR-----EG-----
X_leavis_TRF1  1  -----
G_gallus_TRF1  1  -----MSEA-----GR-----
Danio_rerio  1  ---MAGTFGC-----
Fukomys_damaren  1  ---MADDSSSG-----VQSP-----RG---SA-----
Ustilago_maydis  1  ---MSASARSSTRRTRNSIAAASSSIAASRTRAR-----SASRRQSTTTDSP
Ceraceosorus_bo  1  ---MSTRSSR-----RTNPA-----S-----
Tilletia_caries  1  -----MSRRTHFAQPAP
Puccinia_striif  1  ---MVNKQOSS-----RTSS-----R-----
Rhodotorula_tor  1  ---MSGLSPTK-----RVTS-----S-----
Tilletia_indica  1  ---MTATTYCS-----DSSSEISIHSTRKMPRRAEFAPSAP
consensus    1  .
  
```

fungi

```

Hs_TRF1      17  -----D-----GRDADPT-
Mm_TRF1      18  -----W-TDS-
X_leavis_TRF1  1  -----
G_gallus_TRF1  7  -----
Danio_rerio  8  -----
Fukomys_damaren  17  -----D-----GEDEDAR-
Ustilago_maydis  45  SATQSRTRNASRPQSITASLSAATSQPRTKTTTTRRRNTTGSFQSHYI--VLSSDS
Ceraceosorus_bo  14  -----TSRQAEQGTDLANRSSA
Tilletia_caries  13  MATRSRMQS-L-----RPESGPA-
Puccinia_striif  14  -----L---KNN---QTTSNPSS
Rhodotorula_tor  14  -----S---V---RPLPATAP
Tilletia_indica  34  MGTRSRMQS-L-----RPQAGPS-
consensus    56
  
```

fungi

```

Hs_TRF1      25  -----E---EQMAETERNDEEQFEC-----
Mm_TRF1      22  -----D---SPEQEEVGD-----D-----
X_leavis_TRF1  1  -----
G_gallus_TRF1  7  -----
Danio_rerio  8  -----
Fukomys_damaren  25  -----E---PALIE-DRDNQEHIEC-----
Ustilago_maydis  98  SDSSDSPDSPDSPDSPDSPDSPSPGSPNPSESPVSDSDDDQATPEASDFDYL
Ceraceosorus_bo  31  QAESVIRIRSSSSSNRADSVASSVSTVRPRSKRIQEGPTA-SKRQA--EVKGVGGL
Tilletia_caries  30  -----
Puccinia_striif  26  SA-----ET-----
Rhodotorula_tor  24  RP-----QP-----
Tilletia_indica  51  -----
consensus    111
  
```

fungi

```

Hs_TRF1      42  -----QELL---
Mm_TRF1      33  -----AELL---
X_leavis_TRF1  1  -----
G_gallus_TRF1  7  -----
Danio_rerio  8  -----GFIIINL
Fukomys_damaren  41  -----QKLL---
Ustilago_maydis  153  DLSDASDSHSDPSDASGASDTSDEHQDPSNTLRSNAPRAARQDRSLRESTDDV
Ceraceosorus_bo  83  KIEDEILLE-EDERDLGGEGVEMDV-----QVVETPRPRGTRAPRPTNVVLSFF
Tilletia_caries  30  -----
Puccinia_striif  30  -----TNHEQ-
Rhodotorula_tor  28  -----TAPRMF
Tilletia_indica  51  -----
consensus    166
  
```

fungi

Hs_TRF1	46	-----ECQVQVG-APEE-----
Mm_TRF1	37	-----QCQLQLG-TPRE-----
X_leavis_TRF1	1	-----M-----
G_gallus_TRF1	7	-----E-----
Danio_rerio	15	KME-----SESH-----
Fukomys_damaren	45	-----DCQGEFW-VPEK-----
Ustilago_maydis	208	SSDEAAETSNNVTDAVA-----TSDESQEALDE-ALVASQLTPRPSRHHLES
Ceraceosorus_bo	131	AQDAASSPSSVRGGVGPQIHVFGSSSVRSVEQEAS-RLSGGRLGPPSSQHEVVAS
Tilletia_caries	30	-----YRGM--AGAPRQPSVGMRAPSG-ESST---SIIATLNDNN-----A
Puccinia_striif	35	-----STTAKKIQLGRSTEQTAASSNN--NQTVSNRPK-----N
Rhodotorula_tor	34	T-SANTSPPRSV--GRTPRASLAAVSAPHSVQHNNN--NPSERLSTLA-----T
Tilletia_indica	51	-----YGGI--GVAPRAPSNPVRAPSLAESSMSTIATMERLGDNS-----A
consensus	221	

fungi

Hs_TRF1	57	-----
Mm_TRF1	48	-----
X_leavis_TRF1	2	-----
G_gallus_TRF1	8	-----
Danio_rerio	22	-----
Fukomys_damaren	56	-----
Ustilago_maydis	254	ADSDTTQTALHRLHPQEQRGNRLARASIAGPLVSSR---DTNHTSISTRSRPSKR
Ceraceosorus_bo	185	PRTDG-----GAPASYALGRPRMSSEIPRAARHTSIGVMAGPDRV
Tilletia_caries	65	RKRRR-----VFSGA-----
Puccinia_striif	68	DSFNL-----ASMKT-----
Rhodotorula_tor	77	PAAQL-----AHSSR-----
Tilletia_indica	90	RKRRR-----IFSGE-----
consensus	276	

fungi

Hs_TRF1	57	-----EEE-----
Mm_TRF1	48	-----MEN-----
X_leavis_TRF1	2	-----E-----
G_gallus_TRF1	8	-----R-----
Danio_rerio	22	-----EIT-----
Fukomys_damaren	56	-----EKE-----
Ustilago_maydis	306	ARHSAPLTRTATSQAQSGRSRSDTARSALSNTPASTRRSKRL--NAGIVISDDESS
Ceraceosorus_bo	225	-----PRRRQVSALTSGAELQE-----LMRSGRL-----
Tilletia_caries	75	-----
Puccinia_striif	78	---LKAMIDAFIKFI-----LRTNRDQR--IQSIQEKRLAEDEQITWDA--IF
Rhodotorula_tor	87	---RAPLTPATTSAV-----TRNAQLVRTPATIRAQARLARGGASGWRA--SL
Tilletia_indica	100	-----
consensus	331	

fungi

Hs_TRF1	60	-----EEEDA-----
Mm_TRF1	51	-----AE---
X_leavis_TRF1	3	-----EETDG-----
G_gallus_TRF1	9	-----EGGLV-----
Danio_rerio	25	-----STSDK-----
Fukomys_damaren	59	-----VDKD-----
Ustilago_maydis	359	QESDEGSEEGDEEPDESDDESEETNTITDKIAAQSGRRPRRSSPRSGADLTANT
Ceraceosorus_bo	249	-----SDSTSGHNTVSFLSDARGGDAFRR-----GS--I-ST
Tilletia_caries	75	-----RKVEEPPIASASGIGAGP-----SGS--RFPD
Puccinia_striif	119	SISNEFITIDDDQELDSVQEQEVILKHNLAHLFVMIIE-----NSSKLIQDFSP
Rhodotorula_tor	130	-----THGGADEARRQLE-----ARRGLVQESAVD
Tilletia_indica	100	-----RRPVAPSLASASGLPRSS-----SGL--R--D
consensus	386	

fungi

```
Hs_TRF1          65 GL--VAEA-----E--
Mm_TRF1          53 -L--VAEV-----E--
X_leavis_TRF1    8  PP--FDDT-----A--
G_gallus_TRF1    14 PFLPSALA-----E--
Danio_rerio      30 TT--SQEV---N-----
Fukomys_damaren  63 -L--VAKA-----E--
Ustilago_maydis  414 TV--AQSE---QRLRPSLA-----DPDGR-----LDVPTIEERCAAI
Ceraceosorus_bo  278 RK--SSVG---PRLRPSLM-----KSDGQ-----VHEPSLAQRSNML
Tilletia_caries  100 DV--KHQP---QRLRPSLE-----HGADGA-----LNVQDVIDRCSIL
Puccinia_striif  168 IL--ESLLIYNLQHEKNLIDQSFRFIIIELEGYRFHAWLKHLVHAHSEVTLFNQS
Rhodotorula_tor  155 AL--ERLLMVG----RDGVGEDNGDWTMGED-----
Tilletia_indica  123 AS--NRQP---QRLRPSLE-----FGANGE-----LNAQAVIDRCSIL
consensus       441 .
```

Reviewers' comments:

Reviewer #1 (Remarks to the Author):

The authors made a substantial effort to address the comments raised during the first round of review. As a result their manuscript is, in my opinion, suitable for publication.

Reviewer #2 (Remarks to the Author):

As I fully understand challenges regarding the expression of the full-length variant of UmTrf2, the major conceptual drawback for me is still the fact that the authors performed several experiments with truncated variant of UmTrf2 lacking N-terminal part. In case of human TRF2, N-terminal part is the part that is responsible for additional DNA binding modes. Thus, if only the truncated variant is investigated, we can miss important functional contributions of N-terminal part of UmTrf2. The authors suggested that "(the first 414 amino acids) is absent from metazoans and poorly conserved among fungal species (see the alignment at the end of this file), suggesting that it is not critical for function." I am just wondering why authors omitted hSTRF2, supposed human analogue of UmTRF2, in the Alignment UmTrf2 N-terminus at the end of the Rebuttal Letter.

I am afraid that the conclusions drawn are still insufficiently supported by the data. For example, the authors added only replicates to the previously shown data points (Fig. 1B, 1C, 2B, 2C) without increasing number of new data points as suggested by the Reviewer #2 and the main editor as well.

Even though that the manuscript contains now more reproducible data, the authors increased level of detail provided, methodology information and validity of statistical analyses, it still needs substantial improvements. Therefore, I suggest rejecting the manuscript.

reviewer #3 (Remarks to the Author):

Did not return report.

Reviewer #4 (Remarks to the Author):

The paper by Yu et al. examines two putative telomere binding proteins in *U. maydis* – named UmTrf1 and UmTrf2 – and demonstrates that these two proteins play distinct roles in telomere biology. Notably, both proteins are capable of binding telomeric DNA, but UmTrf1 primarily functions in telomere replication whereas UmTrf2 primarily functions in telomere protection. Strikingly, knockdown of UmTrf1 suppresses a Ku70 mutant similarly to knockdown of Blm1, and knockdown of UmTrf2 results in telomere deprotection similar to knockdown of Ku70. These results are supported by a mixture of in vitro and in vivo assays.

A truncated UmTrf2 is used for in vitro assays designed to test the ability of UmTrf2 to bind to telomeric DNA, but this truncated protein is not a problem for supporting the conclusion that UmTrf2 binds to telomeric DNA. Although ideally these experiments would use a full-length protein, the truncated UmTrf2 nonetheless clearly binds to telomeric DNA, supporting the conclusion drawn by the authors. Moreover, additional claims about the functional relevance of UmTrf2 are based on mutant lines rather than biochemical work with the truncated variant. Thus, sufficient evidence is provided to support the claims about the role of UmTrf2 in telomere protection in this study.

While the results are interesting and important contributions to the field, there are a few details of the paper that need to be addressed:

Major issues

1. The overall logical progression is confusing due to the nomenclature of the proteins under investigation. The assumption seems to be that UmTrf1 and UmTrf2 are homologous and at least somewhat redundant. However, they were identified bioinformatically, not genetically, meaning that the underlying assumption must be supported by rigorous experimentation. The experiments performed indicate that these two proteins exhibit distinct roles in telomere biology, belying the original assumption. Moreover, the nomenclature ascribed to these proteins is perhaps misleading. Considering that these proteins are not 1:1 analogs of mammalian TRF1/2, I think that it may be better to provide distinct nomenclature for UmTrf1/2 to avoid confusion among other scientists in the future. UmTrf1 and UmTrf2 should be given distinct nomenclature that more accurately reflects their evolutionary history. Naming these proteins Trf1 and Trf2 invites the reader to compare these proteins directly to mammalian TRF1 and TRF2, but such direct comparisons do not seem appropriate and will lead to confusion. Perhaps UmTrf1 should be called UmTay1 whereas UmTrf2 should be called UmTbf1.

2. The claim on p. 6 that "the two *U. maydis* proteins evidently recognize telomere repeat DNA by the same mechanism" is insufficiently supported. Is there a highly similar domain shared by these proteins that may explain this mechanism? How similar are the Myb domains? An alignment is not shown. To support this claim, the hypothesized domains should be deleted/mutated to prevent binding using the same *in vitro* assay that shows these proteins recognize telomere repeat DNA.

3. The alignment shown in Supp. Fig. 1 needs more annotations. Insertions are indicated in the alignment but not the conserved/catalytically active residues. Which residues are expected to be critical in the Myb domain for binding telomere repeat DNA? Such residues should be indicated.

4. No data are shown confirming the mutations. The method section says "correct mutant strains were identified by PCR analysis" but does not provide any details. Primers used for this analysis should be clearly indicated in a supplemental table. Additionally, western blot or qPCR analysis are needed to demonstrate that the targeted gene has been knocked out/down. Fig. 3A and Fig. 4D&E in particular highlight the need for these data as the clones have different phenotypes that might be explained by differential knockdown.

Minor issues

1. There are a few extraneous commas that should be removed.

a. On p. 3: "repair events at the termini, and by promoting"

b. On p. 3: "consecutive Myb motifs away from the C-terminus, and evidently functions"

c. On p. 11: "regulates telomere recombination, but is not essential"

d. On p. 12: "function in ALT-related pathways, and could have relevance"

2. There is a missing comma. On p. 4, the sentence beginning "Trf2 is evidently essential and transcriptional repression of this gene..." needs a comma after "essential".

3. In the introduction, "TRF" and "TRFH" should be defined on their first usage ("Telomere Repeat Binding Factor" and "TRF Homology")

4. On p. 5, it is stated that "both insertion 2 and 3 are relatively invariant in length and contain well conserved residues" but this claim is not clear. Compared to what are the insertions invariant in length?

5. On p. 6, in the sentence beginning "To test the DNA-binding activity of UmTrf2 directly..." replace "was unsuccessful" with "were unsuccessful."

6. The section header on p. 8 "trf2 is crucial for telomere protection" should include "Trf2" rather than "trf2" to be consistent with the other section headers.

7. In the discussion on p. 11, the phrasing is unclear in which organism "tay1Δ haploid strains could not be generated" – *S. pombe* or *Y. lipolytica*.

8. In the same sentence on p. 11, the em dash should be replaced with a semi colon.

9. The legend for Figure 1A should include the adjective "recombinant": "All of the *U. maydis* recombinant proteins examined..."

10. Fig. 4A and Fig. 4B should not have different scales on the y-axes as this is a bit misleading and

exaggerates the result in Fig. 4A. Additionally, the rationale for using 12.5 nM or 25 nM protein in these two experiments should be included somewhere in the text.

Reviewer #1 (Remarks to the Author):

The authors made a substantial effort to address the comments raised during the first round of review. As a result their manuscript is, in my opinion, suitable for publication.

We thank the reviewer for the positive assessment.

Reviewer #2 (Remarks to the Author):

As I fully understand challenges regarding the expression of the full-length variant of UmTrf2, the major conceptual drawback for me is still the fact that the authors performed several experiments with truncated variant of UmTrf2 lacking N-terminal part. In case of human TRF2, N-terminal part is the part that is responsible for additional DNA binding modes. Thus, if only the truncated variant is investigated, we can miss important functional contributions of N-terminal part of UmTrf2. The authors suggested that “(the first 414 amino acids) is absent from metazoans and poorly conserved among fungal species (see the alignment at the end of this file), suggesting that it is not critical for function.” I am just wondering why authors omitted hsTRF2, supposed human analogue of UmTRF2, in the Alignment UmTrf2 N-terminus at the end of the Rebuttal Letter.

I am afraid that the conclusions drawn are still insufficiently supported by the data. For example, the authors added only replicates to the previously shown data points (Fig. 1B, 1C, 2B, 2C) without increasing number of new data points as suggested by the Reviewer #2 and the main editor as well.

Even though that the manuscript contains now more reproducible data, the authors increased level of detail provided, methodology information and validity of statistical analyses, it still needs substantial improvements. Therefore, I suggest rejecting the manuscript.

No response per editor’s decision letter.

reviewer #3 (Remarks to the Author):

Did not return report.

Reviewer #4 (Remarks to the Author):

The paper by Yu et al. examines two putative telomere binding proteins in *U. maydis* – named UmTrf1 and UmTrf2 – and demonstrates that these two proteins play distinct roles in telomere biology. Notably, both proteins are capable of binding telomeric DNA, but UmTrf1 primarily functions in telomere replication whereas UmTrf2 primarily functions in telomere protection. Strikingly, knockdown of UmTrf1 suppresses a Ku70 mutant similarly to knockdown of Blm1, and knockdown of UmTrf2 results in telomere deprotection similar to knockdown of Ku70. These results are supported by a mixture of in vitro and in vivo assays.

A truncated UmTrf2 is used for in vitro assays designed to test the ability of UmTrf2 to bind to telomeric DNA, but this truncated protein is not a problem for supporting the conclusion that UmTrf2 binds to telomeric DNA. Although ideally these experiments would use a full-length protein, the truncated UmTrf2 nonetheless clearly binds to telomeric DNA, supporting the conclusion drawn by the authors. Moreover, additional claims about the functional relevance of UmTrf2 are based on mutant lines rather than biochemical work with the truncated variant. Thus, sufficient evidence is provided to support the claims about the role of UmTrf2 in telomere protection in this study.

We thank reviewer 4 for affirming that the biochemical analysis in the paper is adequate to support the conclusions.

While the results are interesting and important contributions to the field, there are a few details of the paper that need to be addressed:

Major issues

1. The overall logical progression is confusing due to the nomenclature of the proteins under investigation. The assumption seems to be that UmTrf1 and UmTrf2 are homologous and at least somewhat redundant. However, they were identified bioinformatically, not genetically, meaning that the underlying assumption must be supported by rigorous experimentation. The experiments performed indicate that these two proteins exhibit distinct roles in telomere biology, belying the original assumption. Moreover, the nomenclature ascribed to these proteins is perhaps misleading. Considering that these proteins are not 1:1 analogs of mammalian TRF1/2, I think that it may be better to provide distinct nomenclature for UmTrf1/2 to avoid confusion among other scientists in the future. UmTrf1 and UmTrf2 should be given distinct nomenclature that more accurately reflects their evolutionary history. Naming these proteins Trf1 and Trf2 invites the reader to compare these proteins directly to mammalian TRF1 and TRF2, but such direct comparisons do not seem appropriate and will lead to confusion. Perhaps UmTrf1 should be called UmTay1 whereas UmTrf2 should be called UmTbf1.

The reviewer's point is well taken. We have switched the designation of UmTrf1 to UmTay1 in order to underscore the structural distinction of this protein family from mammalian TRFs. In contrast, we have retained the UmTrf2 designation because of its structural and function similarity to mammalian TRF2 (especially in light of the shared telomere protection function). We believe the revised nomenclature is appropriate from both the evolutionary and functional perspectives.

2. The claim on p. 6 that “the two *U. maydis* proteins evidently recognize telomere repeat DNA by the same mechanism” is insufficiently supported. Is there a highly similar domain shared by these proteins that may explain this mechanism? How similar are the Myb domains? An alignment is not shown. To support this claim, the hypothesized domains should be deleted/mutated to prevent binding using the same *in vitro* assay that shows these proteins recognize telomere repeat DNA.

UmTay1 and *UmTrf2* each contain Myb domains that are similar to the equivalent domain in mammalian TRFs, and that can explain the similarities in DNA-binding mechanisms. We have now provided alignments that highlight the DNA binding residues of both family of fungal proteins, which are well conserved and shared by the mammalian TRFs (Supp. Fig. 1B and 1C). In terms of experimental validation, we have shown in the previous version of the manuscript that the Myb domains of *Tay1* is sufficient for high affinity binding to telomere DNA (*Tay1*^{N274}, Fig. 1). In this revision, we further analyzed the DNA binding activity of the TRFH (dimerization) domain and the Myb domain of *UmTrf2* (Supp. Fig. 2B). As expected based on earlier studies of mammalian TRFs, only the Myb domain exhibited appreciable DNA-binding, and the binding affinity of this domain is substantially reduced relative to *Trf2ΔN* owing to the inability to dimerize (Supp. Fig. 2A and 2B).

3. The alignment shown in Supp. Fig. 1 needs more annotations. Insertions are indicated in the alignment but not the conserved/catalytically active residues. Which residues are expected to be critical in the Myb domain for binding telomere repeat DNA? Such residues should be indicated.

The residues are now indicated. See the response to comment #2.

4. No data are shown confirming the mutations. The method section says “correct mutant strains were identified by PCR analysis” but does not provide any details. Primers used for this analysis should be clearly indicated in a supplemental table. Additionally, western blot or qPCR analysis are needed to demonstrate that the targeted gene has been knocked out/down. Fig. 3A and Fig. 4D&E in particular highlight the need for these data as the clones have different phenotypes that might be explained by differential knockdown.

We have now included the genotyping data for individual *tay1Δ*, *trf2^{arg1}*, and *ku70^{nar1} tay1Δ* mutant strains (the strains analyzed in Fig. 3A, 4D, 4E, 5C and 6A) along with the primers used for these analyses (Supp. Fig. 12 and Supp. Table S2). Each strain was verified by PCR reactions that tested both the disruption of the endogenous locus

and the integration of the cassette into the correct chromosomal locus. Antibodies for these proteins are unavailable, and given the genotyping results, there does not appear to be compelling reasons for performing Western or qPCR analysis. We also note that despite some variations in the phenotypic severities of mutants of the same genotypes (e.g, the degree of telomere shortening in the *tay1Δ* mutants), the key conclusions concerning the telomeric functions of Tay1 and Trf2 are supported by each mutant clone.

Minor issues

1. There are a few extraneous commas that should be removed.

a. On p. 3: “repair events at the termini, and by promoting”

b. On p. 3: “consecutive Myb motifs away from the C-terminus, and evidently functions”

c. On p. 11: “regulates telomere recombination, but is not essential”

d. On p. 12: “function in ALT-related pathways, and could have relevance”

Corrected.

2. There is a missing comma. On p. 4, the sentence beginning “Trf2 is evidently essential and transcriptional repression of this gene...” needs a comma after “essential”.

Corrected.

3. In the introduction, “TRF” and “TRFH” should be defined on their first usage (“Telomere Repeat Binding Factor” and “TRF Homology”)

Defined.

4. On p. 5, it is stated that “both insertion 2 and 3 are relatively invariant in length and contain well conserved residues” but this claim is not clear. Compared to what are the insertions invariant in length?

Compared to Insertion 1. This is now clarified in the text.

5. On p. 6, in the sentence beginning “To test the DNA-binding activity of UmTrf2 directly...” replace “was unsuccessful” with “were unsuccessful.”

Corrected.

6. The section header on p. 8 “trf2 is crucial for telomere protection” should include “Trf2” rather than “trf2” to be consistent with the other section headers.

Corrected.

7. In the discussion on p. 11, the phrasing is unclear in which organism “*tay1Δ* haploid strains could not be generated” – *S. pombe* or *Y. lipolytica*.

Clarified.

8. In the same sentence on p. 11, the em dash should be replaced with a semi colon.

Corrected.

9. The legend for Figure 1A should include the adjective “recombinant”: “All of the *U. maydis* recombinant proteins examined...”

Clarified.

10. Fig. 4A and Fig. 4B should not have different scales on the y-axes as this is a bit misleading and exaggerates the result in Fig. 4A. Additionally, the rationale for using 12.5 nM or 25 nM protein in these two experiments should be included somewhere in the text.

Revised as recommended.

REVIEWERS' COMMENTS:

Reviewer #4 (Remarks to the Author):

The authors have made substantial revisions to the manuscript and provided additional experimental evidence that satisfactorily address concerns raised in the previous round of review. The new sentence at the end of the abstract ("Our findings also point to potential stimulatory effect of telomere proteins on ALT in cancer cells.") is too bold of a claim for an abstract. No experiments were performed using cancer cells, so such speculation ought to be limited to the discussion section of the article. Replacing the phrase "in cancer cells" to something such as "in *U. maydis* that may be conserved in other systems" is more appropriate.

Aside from the slight concern regarding the abstract, the article is suitable for publication in my opinion.

Reviewer #4 (Remarks to the Author):

The authors have made substantial revisions to the manuscript and provided additional experimental evidence that satisfactorily address concerns raised in the previous round of review. The new sentence at the end of the abstract ("Our findings also point to potential stimulatory effect of telomere proteins on ALT in cancer cells.") is too bold of a claim for an abstract. No experiments were performed using cancer cells, so such speculation ought to be limited to the discussion section of the article. Replacing the phrase "in cancer cells" to something such as "in U. maydis that may be conserved in other systems" is more appropriate.

Aside from the slight concern regarding the abstract, the article is suitable for publication in my

opinion. **We adopted the suggested change.**